# Impact of functional synapse clusters on neuronal response selectivity

Balázs B. Ujfalussy [1✉] & Judit K. Makara[1]

Clustering of functionally similar synapses in dendrites is thought to affect neuronal input-output transformation by triggering local nonlinearities. However, neither the in vivo impact of synaptic clusters on somatic membrane potential (sVm), nor the rules of cluster formation are elucidated. We develop a computational approach to measure the effect of functional synaptic clusters on sVm response of biophysical model CA1 and L2/3 pyramidal neurons to in vivo-like inputs. We demonstrate that small synaptic clusters appearing with random connectivity do not influence sVm. With structured connectivity, ~10–20 synapses/cluster are optimal for clustering-based tuning via state-dependent mechanisms, but larger selectivity is achieved by 2-fold potentiation of the same synapses. We further show that without nonlinear amplification of the effect of random clusters, action potential-based, global plasticity rules cannot generate functional clustering. Our results suggest that clusters likely form via local synaptic interactions, and have to be moderately large to impact sVm responses.

---

[1] Laboratory of Neuronal Signaling, Institute of Experimental Medicine, 1083 Budapest, Hungary. ✉email: balazs.ujfalussy@gmail.com

Processing of synaptic stimuli targeting the dendritic tree fundamentally depends on the spatio-temporal structure of the inputs: spatially distributed or asynchronous inputs are integrated linearly, whereas spatially close and synchronous inputs can induce dendritic nonlinearities, such as regenerative dendritic spikes. These observations motivated the idea of functional synaptic clustering: to elicit dendritic spikes, inputs showing correlated in vivo activity should target nearby dendritic locations[1]. Consistent with this idea, in vivo imaging of the activity of dendritic spines demonstrated that neighbouring synapses are co-activated more often than random[2–6] suggesting the involvement of active processes in the formation of functional clusters. However, both the relative importance of synaptic clustering compared to other factors influencing neuronal responses under in vivo conditions and the biophysical mechanisms leading to their formation are unknown.

In particular, the spatial scale of the synapses showing correlated activity in vivo has been found to be restricted to ~5–10 μm and involved a small number, ~2–5 dendritic spines[2,4–6]. This is substantially less than the ~10–20 inputs required to trigger dendritic Na$^+$ or NMDA spikes (characteristic for thin branches) under in vitro conditions[7–10] leaving the potential impact of the clusters elusive. The high background activity, characteristic for the in vivo states, can markedly change the integrative properties of the cell[11–13], but it is not clear how it influences the effect of functional clustering, i.e., whether the facilitation of dendritic spikes[14], or other effects, such as saturation[15], shunting[7,16] or increased trial-to-trial variability[17] are stronger.

The formation of small functional synapse clusters likely involves activity-dependent synaptic plasticity mechanism(s)[18,19]. Synaptic plasticity is influenced by both global (i.e., cell-wide) and local (i.e., dendritic) processes, and how these factors interact to generate functional clustering is not known. On one hand, several lines of evidence indicate that clusters can be generated through local plasticity mechanisms. First, synaptic plasticity has been shown to be driven by local synchrony of the inputs[3]. Second, local cooperative synaptic plasticity mechanisms have been described in dendrites acting independently of the somatic output of the neuron on the spatial scale of synaptic clustering, i.e., 5–10 μm[20,21]. Finally, functional clustering was not restricted to inputs showing correlated activity with the soma[5]. On the other hand, theoretical considerations argue[22,23] and in vivo experimental evidence demonstrates[24] that the somatic output of the neuron influences plasticity of the incoming synapses. Global plasticity may strengthen functional synaptic clustering if co-activation of clustered synapses facilitates somatic action potentials that, back-propagating to the dendritic tree, can reinforce these synaptic clusters. Importantly, this global scenario is expected to require the synaptic clusters to control global synaptic plasticity by driving the output of the cell via the amplification of their postsynaptic effect by local dendritic nonlinearities. Global plasticity can not only strengthen existing clusters but may also contribute to the formation of the synapse clusters if even randomly occurring, small synaptic clusters can trigger dendritic nonlinearities.

In this paper we first demonstrate that global plasticity mechanisms can indeed lead to functional synaptic clustering when synaptic clusters can influence the somatic response. Next we develop a novel analysis method to estimate the effect of synaptic clustering on the somatic response of a biophysical model neuron under in vivo-like conditions. Using our method we show that, when the connectivity is unstructured, small synaptic clusters do not influence the sVm response of CA1 and L2/3 pyramidal neuron models and thus global plasticity can not be responsible for the formation of synaptic clusters in these cell types. We further show that assuming uniform synaptic strength,

10–20 synapses per cluster are required to achieve reliable output tuning, but changing the strength or the number of inputs has a stronger effect on the neuronal output. Finally we demonstrate that the increase of the background activity during hippocampal sharp waves (SPW) paradoxically decreases the effect of synaptic clustering on the sVm, which effect was partially alleviated when the clusters innervated strongly excitable dendritic branches.

## Results

**Clustering via global plasticity needs local nonlinearities.** To examine the theoretical conditions of creating functional synaptic clusters via global mechanisms, we first turned to a simplified neuron model equipped with 10 nonlinear subunits, corresponding to idealised dendritic branches (Methods, Fig. 1a) and simulated structural synaptic plasticity in the model. The model received 100 synaptic inputs from 10 functional presynaptic ensembles with winner-take-all dynamics, representing groups of place cells with non-overlapping place fields. The plasticity of synapse $i$ was controlled by a synapse-specific factor $\phi_i(t)$ representing biochemical processes stabilising the synapse. We changed $\phi_i(t)$ depending on the synaptic input $s_i$ and the global output of the cell, $r(t)$:

$$\Delta\phi_i(t) = r(t)(\alpha\, s_i(t) - \beta) \qquad (1)$$

where $\alpha$ and $\beta$ are parameters controlling the amount of potentiation and depression, respectively. When $\phi_i(t) \leq 0$ (i.e., when the synapse was typically not coactive with the soma), the synapse was replaced by an other input selected randomly from the presynaptic ensembles (Methods). By changing the shape of the subunit nonlinearity (Fig. 1b) we could vary the contribution of local nonlinear integration to the neuronal response (Fig. 1c).

We then simulated plasticity starting from a random initial connectivity between the presynaptic ensembles and the subunits, and compared the developing innervation patterns with a shuffled control. As expected, when subunit nonlinearities were weak, and thus randomly occurring synaptic clusters did not have an additional contribution to the neuronal output (Fig. 1c, orange), the connectivity pattern remained random (Fig. 1d, e, orange). However, when the effect of accidentally formed clusters was boosted by strong subunit nonlinearities (Fig. 1b, green), and thus subunit nonlinearities contributed significantly to the neuron's output (Fig. 1c, green) the connectivity pattern became significantly non-random (Fig. 1d, green): inputs from one ensemble became dominant within a subunit at the expense of other ensembles. This heterogeneity in the input tuning was reflected by the increased variance of the ensemble size within branches compared with shuffle control (Fig. 1e). Thus, the same global plasticity process led to the formation of functional synaptic clusters when subunit nonlinearities were strong.

These simulations demonstrated that functional clustering can be achieved or reinforced via global plasticity mechanism when local nonlinearities contribute to the neuronal response variability. In the following sections we investigate whether local dendritic nonlinearities in cortical pyramidal neurons are sufficiently strong to amplify the effect of input clusters and to control the neuronal responses under realistic input conditions.

**Random synaptic clusters have no impact on somatic responses.** Even with unstructured connectivity, co-tuned synapses may be located near each other in dendrites by chance. First, we measured the impact of such randomly occurring synapse clusters to trigger dendritic nonlinearities and to influence the somatic response. To estimate the impact of functional synaptic clustering under in vivo-like conditions in neurons with complex morphological and biophysical properties, we developed a novel

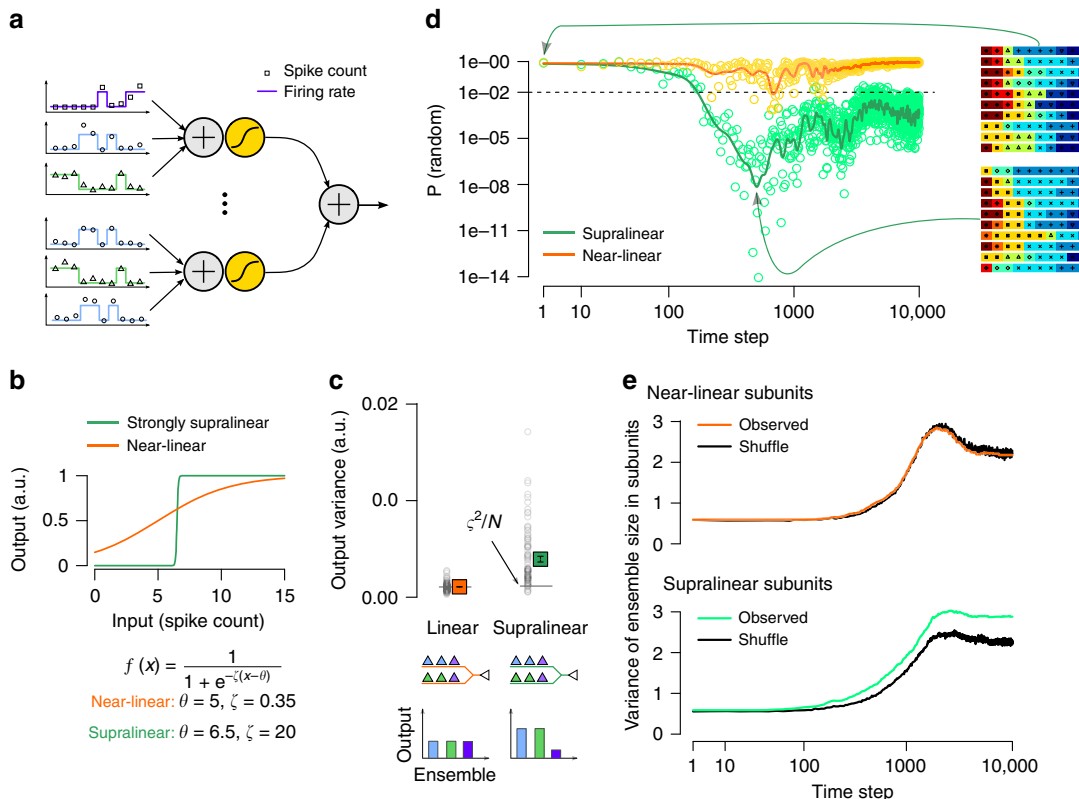

**Fig. 1 Global plasticity can lead to functional clustering. a** Input integration in a simplified neuron model. Left: Presynaptic inputs ($N = 100$, only 6 shown) divided into 10 discrete ensembles (colours; 3 shown). The firing rate (lines) of the ensembles switched between a low and a high activity state, and the activation of a particular ensemble corresponded to a specific spatial location. Spike counts (symbols) were sampled from a Poisson distribution in 100 ms bins, corresponding to theta cycles. Right: Each of the 10 dendritic subunits (2 shown) first integrated the incoming spike counts ($+$ sign) and then applied a pointwise nonlinearity (yellow sigmoidal function) modelling dendritic spikes. The output of the neuron was the sum of the inputs from all branches. **b** By changing the parameters of the sigmoid subunit nonlinearity we could interpolate between near-linear (orange) and strongly supralinear (green) integration. **c** Variance of the postsynaptic tuning curve (mean response to the activation of the 10 ensembles) with near linear and supralinear subunits. Symbols show mean and SEM of $N = 100$ input connectivity, horizontal line indicates the variance expected from trial-to-trial variability, $\varsigma^2/N$ (Methods). Bottom: schematics illustrating the connectivity of the inputs and the output of the cell when the same number of inputs is selected from each ensemble (uniform input condition, Methods) with 2 subunits and 3 ensembles. **d** The probability that the observed connectivity pattern is consistent with random innervation with linear (yellow) and supralinear (green) subunits. The connectivity becomes significantly non-random after $\approx 200$ time steps in the case of supralinear subunits (green). Note the logarithmic $x$ and $y$ axis. Green and orange lines indicate moving average. Insets show the identity of the presynaptic ensembles (colours and symbols) targeting the 10 dendritic subunits (horizontal rows) in one typical neuron at the beginning of the simulations (top) and at the most clustered stage (bottom). **e** Variance of the cluster size was larger than in the shuffled control in the model with supraliner subunits, indicating the presence of functional clustering.

analysis termed the decomposition of response variance (Methods). In short, we simulated a biophysical model neuron whose integrative properties have been fitted to in vitro data, and stimulated it with input patterns matched to the input the neuron receives under behaviourally relevant in vivo conditions via excitatory and inhibitory synapses distributed throughout the entire dendritic tree. We recorded and analysed the biophysical model's sVm response while manipulating the variability of the input and the fine-scale arrangement of the synapses.

We used a detailed model of a CA1 pyramidal cell[25] (Methods) that reproduced several somatic and dendritic properties of these neurons measured under in vitro conditions (Supplementary Fig. 1a–g), including the generation and propagation of Na$^+$ action potentials at the soma and along the apical dendritic trunk[25]; the generation of local Na$^+$ spikes in thin dendritic branches[8]; amplitude distribution of synaptic responses[26]; nonlinear integration of inputs via NMDA receptors[8] (NMDAR); the similar voltage threshold for Na$^+$ and NMDA nonlinearities[8] and the major role of A-type K$^+$ channels in limiting dendritic excitability[8,27].

After validating the biophysical model, we stimulated the neuron with synaptic input patterns characteristic for the hippocampal population activity under natural conditions (Methods). Specifically, we simulated the activity of 2000 excitatory and 200 inhibitory presynaptic neurons during the movement of a mouse in a 2-m long circular track (Fig. 2a, b). The excitatory neurons exhibited a single idealised place field, were modulated by theta oscillation and showed phase precession[28] (Supplementary Fig. 1h, i). As about 10% of hippocampal CA3 neurons show location selective activity in a given environment[29,30], the activity of the simulated cells accurately represents the CA3 inputs received by the a postsynaptic CA1 neuron under in vivo conditions (Methods). Inhibitory interneurons were modulated by theta oscillation[31] but were spatially untuned[32]. Importantly, the biophysical model showed place-selective activity in response to the synaptic inputs, with several features of the sVm activity falling in the physiological range[33] (Supplementary Fig. 1j–l). Initially we chose unstructured connectivity between the inputs and the postsynaptic dendritic tree and studied how the sVm response of the neuron was

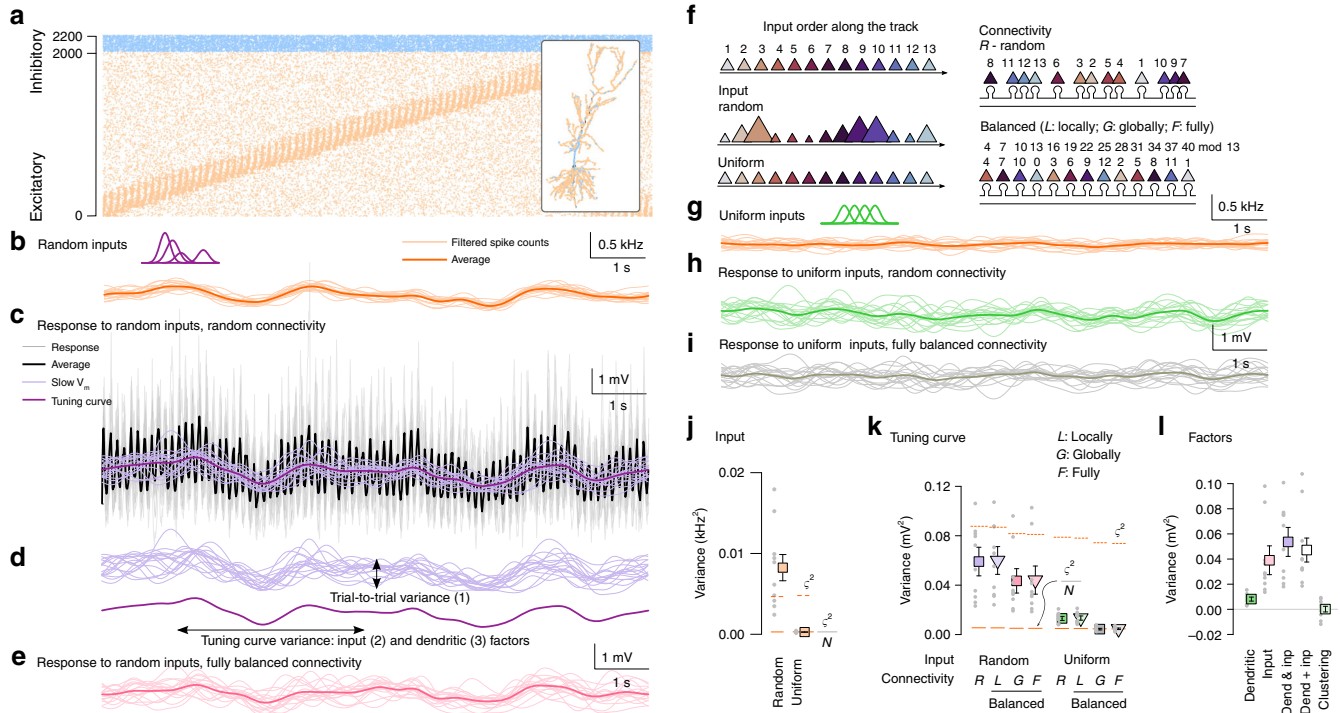

**Fig. 2 Random synaptic clusters have small impact on neuronal tuning.** Source data are provided as a Source Data file. **a** Activity of the 2000 excitatory (orange, ordered by the place field location) and 200 inhibitory (blue) inputs in a single lap on the circular track. Inset: Morphology of the modelled CA1 pyramidal neuron and spatial distribution of excitatory and inhibitory synapses. **b** Filtered input spike counts in 16 individual trials (light) and average (dark) in the random input condition (purple: schematic place fields). **c** sVm response of the postsynaptic cell to inputs in 16 laps (grey), the average postsynaptic response (black), the low-pass filtered responses (slow $V_m$, light purple) and the tuning curve (dark purple) in the random input condition with random connectivity. **d** Decomposition of the response variance: trial-to-trial variability (top) and the tuning curve variance (bottom). **e** Slow $V_m$ response in individual trials (light) and tuning curve (dark) in the random input condition with fully balanced connectivity. **f** Schematic for the inputs and connectivities. Top: ordering of 13 input place cells along the circular track. Colour difference is proportional to place field distance. Left: schematic of the random and uniform inputs. The size of the symbols correspond to the total input strength at a given location. Right: schematic of random ($R$) and balanced connectivity on a short segment of a single dendritic branch. To eliminate clustering, we used a co-prime ordering procedure either locally within branches ($L$), globally ($G$) or fully ($F$; see Methods; mod is the modulo operator). **g** Same as **b** in the uniform input condition. **h, i** Same as **e** with uniform inputs and random (**h**) or fully balanced (**i**) connectivity. **j** Variance of the average filtered input spike count in random and uniform input conditions. Solid line segments show the lower bound of the variance of the average of 16 laps based on trial-to-trial variability ($\varsigma^2/N$) and dashed segments indicate trial-to-trial variance. **k** Tuning curve variance in the random and uniform input conditions with either random ($R$) or locally ($L$), globally ($G$) or fully balanced ($F$) connectivity. **l** Contribution of dendritic and input factors to response variance. Grey dots in **j**–**l** show $N = 10$ independent simulations with different synaptic configuration and inputs. Symbols and error bars show mean and SEM.

affected by small functional synaptic clusters (typically 2–4 synapses, depending on the definition of the synaptic cluster, Supplementary Fig. 2) occurring randomly under these conditions.

We observed considerable variability of the sVm response both across trials (trial-to-trial variability) and along the track (tuning curve variability; Fig. 2c, d). To evaluate the contribution of synaptic clustering to the total variability, we compared its effect to other components contributing to the total variance of the neuronal response. In particular, the sVm response variability can be attributed to three separate factors (Fig. 2d): (1) trial-to-trial variability associated with stochastic biophysical processes (e.g., spiking and synaptic vesicle release) modelled as a Poisson process here (Methods); (2) tuning curve variability caused by variations in the inputs active along the track, including the precise number of presynaptic neurons with place fields at a given location, their maximal firing rates or their synaptic strengths; (3) tuning curve variability attributed to dendritic factors, i.e., differential spatial distribution of inputs active along the track including small-scale functional clustering or large-scale spatial heterogeneities (e.g., proximal vs. distal dendritic location). Importantly, in our simulations we could separate the effect of

these factors by simple manipulations of the input conditions (i.e., the strength of the synapses and the location and amplitude of the presynaptic place fields; Fig. 2f, left) and the synaptic connectivity (i.e., the spatial arrangement of the synapses along the dendritic tree; Fig. 2f, right).

We first measured the contribution of the input and dendritic factors together on the variance of the tuning curve under random input conditions (Fig. 2b, f, input place field parameters varied within physiological range; Methods), and random connectivity using 10 independent synaptic configuration patterns. We found that the tuning curve varied considerably along the track (Fig. 2d) and we compared this variability to the trial-to-trial variance ($\varsigma^2$) estimated in $N = 16$ trials. The tuning curve variability was consistently larger than the lower bound imposed by the trial-to-trial variability ($\varsigma^2/N$, Fig. 2k, random input, random connectivity; Methods) indicating a significant contribution from inputs and/or dendritic factors.

Second, we eliminated the effect of dendritic factors on the response variability by rearranging synapses throughout the dendritic tree to minimise the correlation between the activity of nearby synapses (balanced connectivity, Fig. 2f, Supplementary Fig. 3 and Methods). In the balanced connectivity the functional

synaptic clusters are equally absent for all differently tuned presynaptic ensembles, while dendritic processing in general is unchanged. In the first step we rearranged synapses only within individual dendritic branches, removing local synaptic clusters but otherwise keeping large-scale, global biases intact (locally balanced connectivity, L). We found that this manipulation did not change the response variance compared to the random connectivity ($p = 0.38$; Wilcoxon signed rank test (W-test), Fig. 2k), indicating that small functional clusters occurring randomly had no effect on the neuronal tuning. Next, we fully rearranged the synapses removing both local clusters and large-scale inhomogeneities (fully balanced connectivity, F). This manipulation slightly, but significantly decreased the tuning curve variance, in spite of the large variability across simulations (Fig. 2e, k, random vs. fully balanced connectivity, $p = 0.03$, W-test). Randomising synapse locations within branches (globally balanced connectivity, G) did not increase the tuning curve variance compared to the fully balanced connectivity (Fig. 2k, global vs. fully balanced connectivity, $p = 0.5$, W-test).

Third, to isolate and directly measure the effect of dendritic factors, we eliminated the impact of input factors by setting the strength of the synapses identical and the total input rate to constant, but keeping the inputs otherwise unchanged. To achieve this we fixed the shape of the place fields and organised them to uniformly tile the environment (uniform input condition, Fig. 2f, g). This way, the variability of the input was minimised (Fig. 2j) and thus all remaining variability in the tuning curve was attributed to dendritic factors. Using random connectivity, we found that, after eliminating the input factors, the variability of the tuning curve decreased substantially but the remaining variability was still larger than $\varsigma^2/N$ (Fig. 2h, k, random connectivity, uniform vs. random input, $p = 0.002$, W-test). We also observed large reduction in the variability between the 10 simulated neurons indicating that the majority of the cell-to-cell variability was caused by the random selection of inputs. Rearranging synapses within individual dendritic branches did not have an effect on the tuning curve variability (Fig. 2k, uniform input, random vs. locally balanced connectivity, $p = 0.77$, W-test). Finally, when uniform input condition was combined with fully balanced connectivity, the tuning curve variance became statistically identical to the variance expected from the trial-to-trial variability (Fig. 2j, uniform input, balanced connectivity vs. $\varsigma^2/N$, $p = 0.77$, W-test), confirming that our manipulations successfully eliminated both input and dendritic factors. These results were qualitatively similar when we analysed the raw membrane potential responses at the expected peak phase of the theta oscillation (data not shown) indicating that our results are not related to sVm processing. We obtained similar result with L2/3 pyramidal neurons using inputs matched to population activity recorded in the visual cortex in vivo (Supplementary Fig. 4a–e).

From these results we conclude that with unstructured connectivity, input factors provide a substantially stronger contribution to the sVm response of CA1 and L2/3 pyramidal cells than dendritic factors. Moreover, we found that among dendritic factors, large-scale biases in the synaptic input tuning have a measurable contribution to the response variability, but small functional synaptic clusters do not influence neuronal responses (Fig. 2l). As the output of the neuron is controlled by these other factors, back-propagating action potentials are not correlated with the activity of the small synaptic clusters and thus, synaptic clusters can not be reliably reinforced via global, Hebbian plasticity (Fig. 1). Our data therefore supports the need of local synaptic plasticity mechanisms with low activation threshold for the formation of synaptic clusters[21].

Next we introduce structured synaptic connectivity (synaptic clusters) and study how they influence the tuning curve of the neuron.

**Larger synaptic clusters can lead to clustering-based tuning**. To systematically study the effect of synaptic clustering we chose a subset of excitatory synapses that were co-active in the middle of the track (Fig. 3a) and organised them into clusters of increasing size (Fig. 3b; Methods). Specifically, we increased the cluster size from 1 synapse per cluster (no clustering) to 60 synapses per cluster (maximal cluster size) in 7 discrete steps, and measured the somatic and dendritic membrane potential response of the neuron with different levels of clustering to identical, uniform presynaptic inputs (Fig. 3c top and middle). While we did not observe a detectable effect of clusters consisting of 2–5 synapses, when cluster size reached that of 10 synapses, a depolarisation ramp emerged in the tuning curve at the location where the clustered synapses were active (Fig. 3c, d) increasing the variance of the tuning curve (Fig. 3e, bright symbols) and the response integral (Fig. 3f, bright symbols). Note that the input to the neuron is identical across the different clustering configurations, thus the ramp is the somatic signature of nonlinear dendritic processing of clustered inputs: the tuning is entirely due to input clustering. Further increasing the cluster size to 20 synapses per cluster increased the magnitude of the depolarising ramp, but it also saturated the response (Fig. 3e, f, bright symbols). Omitting dendritic spines (Fig. 3e, f, dashed grey line) did not change the minimal cluster size required to achieve reliable tuning. Similarly, randomly distributing the clustered synapses along the entire dendritic branch only marginally changed the tuning, confirming that the fine-scale arrangement of the synapses within a branch has a minor role in shaping sVm selectivity[8] (Fig. 3e, f).

As synaptic factors play an important role in establishing the tuning of hippocampal place cells[34], we compared the effect of synaptic clustering on the tuning curve with the possible effects of synaptic plasticity. As a simple model of synaptic changes during LTP, we doubled the conductance of the clustered synapses and recorded the tuning curve while changing the level of clustering. We found that the effect of synaptic plasticity was larger than the effect achieved by clustering in general (Fig. 3c, bottom; 3e, f, dark symbols). Similar to the control case, maximal depolarisation of the tuning curve was achieved at intermediate cluster sizes (10–20 synapse per cluster, Fig. 3d, bottom) and further increasing the level of clustering greatly saturated the membrane potential of dendritic branches receiving clustered inputs, decreasing the average magnitude of NMDAR current per synapse. Furthermore, we found that dendritic spines boosted the effect of LTP (Fig. 3e, f, dashed grey line). Paralleling the changes in the somatic tuning, the occurrence of dendritic spikes also increased within the place field in the branches receiving clustered input (Fig. 3g). We obtained similar results in the L2/3 neuron: clusters of 10–20 synapses lead to maximal responses and larger clusters yielded saturation (Supplementary Fig. 4f–j).

Our data show that during place cell activity and theta oscillation, clustering-based tuning emerges in CA1 pyramidal cells only if connectivity is nonrandom and the clusters are relatively large. However, neuronal signal integration is highly dependent on the state of the network[12]. Therefore in the next section we examined the effect of synaptic clustering on the postsynaptic response during hippocampal sharp waves.

**Synaptic clustering has small impact during SPWs**. Hippocampal sharp waves are transient, highly synchronous network states where a large fraction of the local network becomes active[35]. During SPWs, the population activity internally recreates

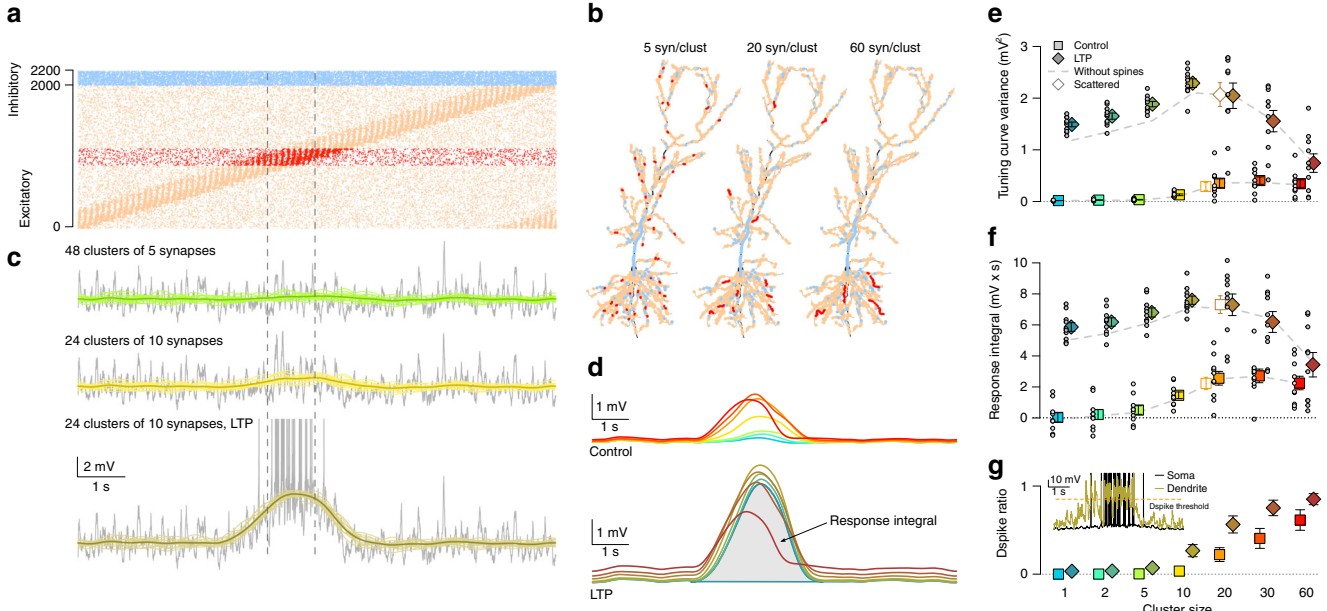

**Fig. 3 Larger synaptic clusters can lead to clustering based tuning.** Source data are provided as a Source Data file. **a** Excitatory inputs (red) coactive in the middle of the track (dashed vertical lines) were chosen for clustered arrangement. **b** Branches longer than 60 µm were selected for arranging 240 synapses (red) into functional synaptic clusters of various size. Examples show clusters of 48 (left), 12 (middle) and 4 (right) synapses. Background excitatory (orange) and inhibitory (blue) synapses are also shown. **c** Example sVm response of the CA1 neuron (grey), slow $V_m$ for 16 trials (green and yellow) and tuning curve (dark green and olive) with 5 (top) or 10 synapses per cluster (middle) or with 10 synapses per cluster combined with LTP of the clustered synapses (bottom). **d** Average of 10 tuning curves with different input configurations shows increased depolarisation ramp amplitude with clustering (top). Potentiation of the clustered synapses further increases the ramp amplitude (bottom). Colour code is the same as in **e**–**g**. **e** Tuning curve variance as a function of synaptic clustering for control (bright squares) and LTP (dark diamonds) in 10 input configurations (circles) and their means (colour boxes) and SEM (error bars). Grey dashed line indicates the mean without dendritic spines. Open symbols indicate simulations when the 20 synapse per cluster was scattered along a the entire branch receiving clustered input. **f** Similar to panel **e** showing the response integral as a function of synaptic clustering. **g** Mean and SEM of dendritic spike ratio within the place field (dashed lines in **a**, **c**) as the function of synaptic clustering across 10 simulations with 4 dendrites each. Inset shows example somatic and dendritic $V_m$.

sequential activity patterns experienced earlier during exploratory behaviour and theta activity[36]. We expected the critical cluster size to be smaller during SPWs than during theta activity, as elevated excitatory activity can facilitate the generation of dendritic spikes[14].

We simulated replay of the experienced trajectory, embedded in an elevated background activity mimicking hippocampal population activity during SPW-ripples (Fig. 4a). The statistics of the excitatory and inhibitory inputs, including the population firing rates, the ripple modulation of the cells and the statistics of the replay events were matched to in vivo data[37–39] (Methods). We systematically varied the overlap between the replayed trajectory with the place fields of the neurons giving postsynaptically clustered inputs (Fig. 4a, b). Contrary to our expectations, we found that the number of action potentials during individual sharp wave events (Fig. 4c, solid lines) and the amplitude of the somatic depolarisation (Fig. 4e, bright symbols) were insensitive of the arrangement of the synapses into functional clusters, although the time spent above dendritic spike threshold in dendritic branches receiving clustered input increased with cluster size (Fig. 4g, bright symbols). Increasing the conductance of the selected excitatory synapses by a factor of 2 (mimicking LTP) resulted in both larger dendritic (Fig. 4d, g) and somatic (Fig. 4e, dark symbols) depolarisation and also a larger output spike count (Fig. 4c, dashed lines) when clusters were small. However, when LTP was combined with large functional synaptic clusters, the advantage of the LTP began to disappear and the output spike counts and somatic depolarisation amplitudes were markedly reduced (Fig. 4c, e). Interestingly, the timing of the action potentials during the SPW correlated with the replayed

input trajectory when the average postsynaptic firing rate was sufficiently high (Supplementary Fig. 5).

To understand the biophysical mechanisms underlying the paradoxical effects of functional clustering during SPWs, we analysed how the local dendritic depolarisation and the average NMDAR current of a single synapse within a functional cluster changes with the size of the cluster (Fig. 4d). We could identify two different mechanisms that contributed to the reduced impact of clustering during sharp waves. First, local dendritic voltage was substantially more depolarised during SPWs than during theta (Fig. 4d, top). Although clustering further increased the local depolarisation during SPWs, it had only a minor impact on the average NMDAR current per synapse (Fig. 4d, bottom) as that was already near-maximal even in the absence of clustering (i.e., cluster size = 1). Second, the elevated synaptic conductance load during SPWs reduced the gain of the neuronal responses such that similar changes in the input currents had a markedly smaller effect on the postsynaptic responses[7] (Fig. 4h).

These results demonstrate that the impact of synaptic clustering on the neural response is small during SPWs and can even be negative when large clusters are combined with LTP and synaptic saturation becomes the dominant effect.

**Strong Na⁺ spikes can overcome synaptic saturation during SPWs.** Regenerative dendritic $Na^+$ spikes can overcome the local synaptic saturation and reduced response gain effects caused by the high input conductance during SPWs and trigger action potentials when propagating to the soma. Local $Na^+$ spikes were relatively weak in our model with only a minor

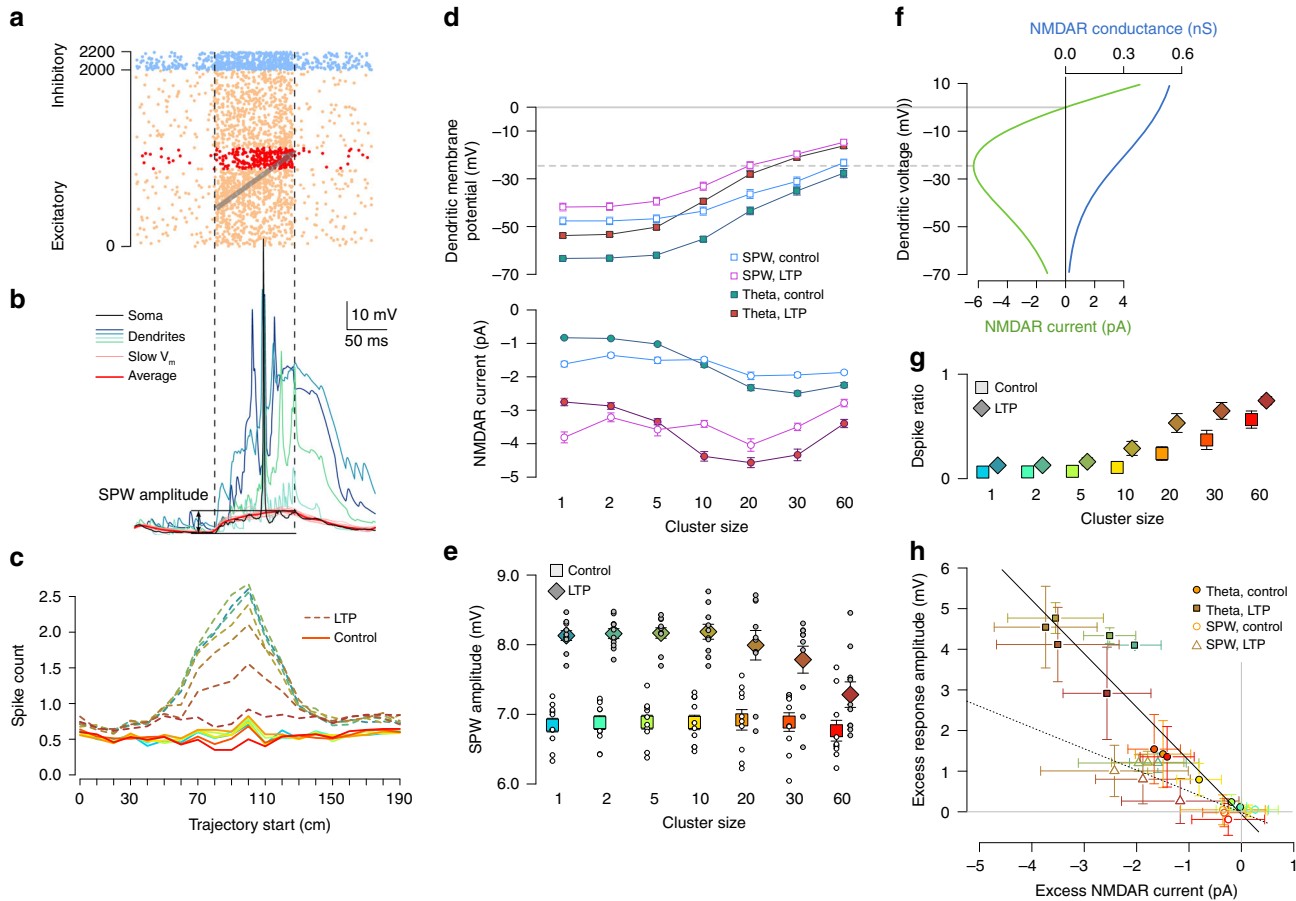

**Fig. 4 Synaptic clustering have reduced impact during sharp waves.** Source data are provided as a Source Data file. **a** Simulated inhibitory (blue) and excitatory (orange and red) inputs during an example SPW (vertical dashed lines). Grey arrow indicates the replayed trajectory. Red dots show activity of the clustered synapses (Fig. 3b). **b** Somatic (black) and dendritic (colours) membrane potential response to SPW inputs in panel **a** (20 synapse per cluster); pink: slow $V_m$ response in 16 trials; red: average sVm of the 16 trials, which was used to measure the amplitude of the depolarisation relative to the pre-SPW period. **c** Average number of somatic action potentials during individual SPW events as a function of the start location of the trajectory for control (solid lines) and LTP (dashed lines). Colour code: same as in panel **e**. The largest responses are when the trajectory starts at 90 and 100 cm, which are analysed further in panels **d**–**h**. **d** Local dendritic voltage at the synaptic clusters (top) and average NMDAR current of clustered synapses (bottom) during the maximal activity of the clustered inputs in SPW events (open symbols) and during theta state (filled symbols). Error bars indicate SEM across 40 dendritic branches (top) or 40 synapses (bottom). **e** Mean somatic depolarisation amplitude relative to the pre-SPW baseline (SPW amplitude, see panel **b**) as a function of cluster size for control (bright squares) and LTP (dark diamonds). Circles show 10 simulations with different cluster arrangements and inputs; symbols and error bars show mean and SEM. **f** NMDAR conductance (blue) and current (green) as a function of voltage. The peak NMDAR current is at ~−25 mV (dashed grey line), which was used as a dendritic spike threshold (panel **g** and Fig. 3g). **g** Mean and SEM of dendritic spike ratio during SPWs as the function of synaptic clustering across 10 simulations with 4 dendrites each. **h** Excess response amplitude in the soma (relative to the one synapse per cluster configuration) as a function of the average NMDAR current during theta (filled circles) and SPWs (open circles). Symbols show mean and SD across 10 cluster arrangements (vertical) or 40 synapses (horizontal). The high variability is explained by the large diversity between dendritic branches receiving clustered inputs.

contribution to the somatic responses (Fig. 5a and Supplementary Fig. 6a). However, experimental evidence shows that a fraction of perisomatic dendrites of CA1 pyramidal cells expresses strong dendritic Na$^+$ spikes that can efficiently promote and time somatic AP firing[40]. Therefore we increased the excitability of selected dendritic branches and studied the effect the strong dendritic spikes generated in these branches on the somatic response during theta and SPWs. Specifically, we selected $N_{strong} = 12$ terminal branches longer than 60 μm, and, as a simple proxy for generating stronger spikes, added a hotspot of voltage gated Na$^+$ channels by increasing the sodium conductance 10-fold in the middle 30% of these branches (Supplementary Fig. 6a). As a result, these branches generated strong dendritic Na$^+$ spikes ( >2V/s dV/dt spike amplitude when measured at the soma[40]) in response to local synaptic stimulation (Fig. 5a).

We found that the presence of strongly excitable dendritic branches had only a minimal effect on the subthreshold responses during theta (Supplementary Fig. 6b, c) although the firing rate of the cell increased slightly in the presence of LTP when clusters were large (Supplementary Fig. 6d). However, strong branches increased both the somatic membrane potential depolarisation (Fig. 5c, top) and the output spike count (Fig. 5d, top) during SPWs when they received input from large ($N_{syn} \geq 20$) synaptic clusters, especially when increased excitability was combined with LTP (Fig. 5c, d, bottom). The coding of the input trajectory by spike timing was similar with strong branches to that in control conditions (Supplementary Fig. 5c–e).

In conclusion, the presence of dendritic branches with strong Na$^+$ spikes made the CA1 model neuron more sensitive to clustering during sharp waves, but still relatively large input clusters were required to make an impact.

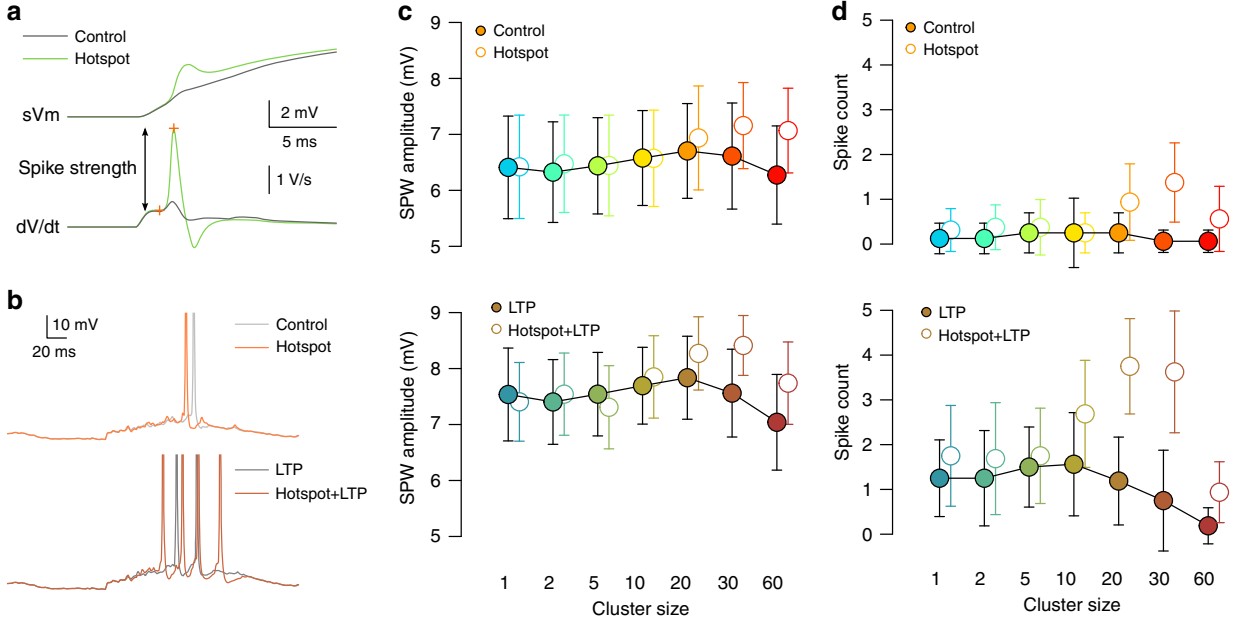

**Fig. 5 The effect of strong dendritic branches on clustering based tuning during SPW activity.** Source data are provided as a Source Data file. **a** Somatic membrane potential (top) and its time derivative (bottom) in response to the activation of 30 synapses in control condition (grey) and with a Na+ hotspot in the middle of the stimulated branch (green). **b** Example responses in control conditions (grey) and with strong dendritic branches (coloured) without (top) and with LTP (bottom) in the case of 10 synapse per cluster when the replayed trajectory maximally overlapped with the place fields of the clustered synapses (trajectory start = 80 cm). Action potentials were truncated. **c** Average somatic Vm depolarisation amplitude during the SPW in control conditions (filled circles) and with strong dendritic branches (empty symbols) as a function of cluster size without (top) and with LTP (bottom) with trajectory start = 80 cm. The presence of strong branches increases the depolarisation and the spike count when the synaptic clusters are large. Symbols show mean and SD of 16 simulations with different inputs but identical cluster arrangements. **d** Similar to **c**, but the average spike count during the SPW is shown.

## Discussion

Identifying the external sensory and internal biophysical correlates of neuronal tuning has been a longstanding goal in systems neuroscience[41,42]. Previous studies estimating the contribution of dendritic nonlinearities to stimulus selectivity relied on pharmacological manipulations which also influenced synaptic factors[43–45]. Our computational approach offers a novel way to isolate these effects and measure the contribution of dendritic and synaptic factors on the neuronal tuning and compare it to the trial-to-trial variability.

We found that synaptic factors have a dominant role over dendritic factors in determining neuronal selectivity under a wide range of behaviourally relevant in vivo-like input conditions both with random and structured connectivity. To ensure that these factors are correctly estimated, we matched the synaptic conductances[46], the EPSP amplitudes[26] and the dendritic active conductances to experimental data[8,25,40]. Stronger dendritic nonlinearities, such as fully propagating dendritic Na+ spikes in hippocampal interneurons[47] or bistable NMDAR nonlinearities[48,49] could increase the contribution of dendritic factors. Conversely, other factors, such as larger differences in the place field properties, greater variability in the synaptic receptor numbers or correlations between the presynaptic firing rates and the synaptic strength are expected to increase the influence of the synaptic factors. In addition, state-dependent changes in neuromodulation may dynamically regulate the relative contribution of synaptic inputs and dendritic processing in vivo[50]. Finally, weak tuning of the inhibitory interneurons[32,42] could increase the magnitude of the synaptic factors, while precisely timed branch-specific inhibitory input could prevent the generation of dendritic spikes[51] and thus alter the dendritic factors.

Local dendritic nonlinearities, in principle, can fundamentally change the integrative properties of neurons and can also have a profound effect on the computational capabilities of the circuit. However, such a large computational effect requires strong local, branch-specific nonlinearities[52] and a large influence on the global output. This is in contrast with our current and recent findings that the effect of local dendritic nonlinearities on the somatic response of the neuron under in vivo-like conditions is relatively small compared to synaptic factors or global, neuron-wide nonlinearities, at least in CA1 and L2/3 pyramidal cells[13]. On the other hand, our work highlights potentially state-dependent roles for different types of local dendritic non-linearities recruited by clustering. While the effect of clustering was mostly mediated by nonlinear NMDAR current during theta oscillation, Na+ spikes generated by strongly excitable dendritic branches rendered neurons sensitive to clustering during SPWs. This raises the possibility that these complementary forms of dendritic nonlinearities—having different kinetic characteristics and overall effects on AP rate and timing—may be activated under different behavioural states associated with different input conditions[8,53]. Multiplexing parallel channels of signal processing with different spatio-temporal properties within the dendritic tree can be a powerful computational strategy[13], but its characterisation requires further theoretical and experimental investigations.

Using our simple model we demonstrated that functional synaptic clusters can be formed and stabilised via global synaptic plasticity rules only in the presence of strong local dendritic nonlinearities (Fig. 1). Notably, this mechanism did not generate clusters composed of functionally homogenous inputs, as global plasticity stabilises any input irrespective of its location within the dendritic tree that show correlated activity with a functional cluster. The interspersion of inputs with a wide range of

functional properties on individual branches is an important characteristic of both experimental data[5,42,54] and computational models[55] and may thus be a signature of the regulation of structural synaptic plasticity on various spatial scales.

Importantly, to generate functional clusters by global plasticity alone, dendritic nonlinearities have to be sufficiently strong with a low threshold to amplify the effect of small, randomly occurring clusters and to control the plasticity process, whereas clustering via local plasticity mechanisms does not necessarily require dendritic nonlinearities[21]. We found that synaptic clusters randomly occurring within dendritic branches did not contribute to the tuning curve variability and thus global plasticity per se is unlikely to account for the reinforcement of small functional synaptic clusters. However, global plasticity can reinforce already existing clusters once they have grown large enough to trigger nonlinear dendritic integration presumably due to local plasticity mechanisms. Moreover, global plasticity can also contribute to the stabilisation of large scale biases in synaptic tuning properties[4] as these factors did have a measurable contribution to the neuronal tuning curve. Whereas several previous theoretical studies showed that local synaptic plasticity processes could gradually generate synaptic clustering[1,56–58], our study indicates their necessity for the initial formation of synaptic clusters. The complementary role of these iterative structural plasticity processes[59] and fast, single trial learning[34] in the generation of the feature selectivity of place cells requires further investigations.

Under in vitro conditions the induction of dendritic spikes requires the near-synchron activation of a sufficiently large number of inputs[8–10,48], but whether inputs occurring in vivo have the sufficient synchrony to trigger them is debated[60,61]. Although the spatial scale of the functional clusters reported in cortical neurons has been restricted to ~5–10 μm and ~2–5 dendritic spines[2,4–6], current experimental techniques do not allow reliable monitoring of the activity of all synaptic inputs in a given dendritic branch and thus, these studies may underestimate the real number of synapses involved in a given synaptic cluster. Moreover, synapses with selectivity similar to the tuning of a small cluster on the same branch can be equally efficient in generating clustering-based tuning as a single, larger synapse cluster (Fig. 3e, f). These considerations suggest that further improvements in the experimental techniques and analysis methods are required to reliably estimate the size and the somatic effect of in vivo occurring synapse clusters. Furthermore, the temporal window for interaction between inputs targeting neighbouring dendritic spines for synaptic plasticity was often found to be orders of magnitude larger[20,62] (≈10 min) than the synchrony required for nonlinear dendritic integration[8] (<10 ms). Even if the dendritic nonlinearities are too weak to substantially amplify the neuronal responses to clustered inputs or inputs are too asynchronous to trigger dendritic spikes, synaptic clustering could have an important role in reducing the interference between memories and to promote selective generalisation via spatially restricting the effect of plasticity-related molecules[20,62,63].

There are several main areas where our results could influence the current interpretation of experimental data. First, our finding that small synaptic clusters do not influence the neuronal responses challenges the prevailing view that the primary function of synaptic clusters is to trigger dendritic nonlinearities and enable flexible single neuron representations[24–6,18,52,64]. Second, our result that global plasticity can contribute to the weakening of other synapses when the cell is driven by the activation of clustered inputs can promote new theoretical and experimental research investigating the interaction between local and global plasticity rules in shaping neuronal feature selectivity and cluster formation[65]. Third, our insights could motivate novel in vivo

experiments both to further quantify the mid-scale organisation and diversity of synaptic inputs targeting the dendritic tree (focusing less on the fine-scale within-branch arrangements[5,6]), as well as to directly test predictions of our simulations.

One fundamental prediction is that small clusters of synapses have minimal effect on the response of a neuron under in vivo conditions. A direct way to test this prediction would be to stimulate a set of inputs of a neuron in vivo in clustered and in distributed configurations (e.g., by in vivo two photon glutamate uncaging[66]) and compare the resulting somatic response. Another critical insight of our theory is that global plasticity does not account for reinforcement of small coactive synapse clusters. This prediction could be tested by a combination of advanced imaging techniques, whereby one measures the activity of both small functional synapse clusters and the soma[4] and monitors long-term plasticity of the clustered synapses[19]. Specifically, our theory predicts that small clusters of coactive synapses will be strengthened even if they are uncorrelated with somatic activity. While currently both experiments are beyond tractability with available techniques, they could directly test the predictions of our model in the foreseeable future.

In conclusion, our findings indicate that (1) the selective responses of cortical neurons are primarily the consequence of the tuning of their synaptic inputs, (2) functional synaptic clustering matched to local dendritic properties can have additional role in refining those responses, (3) plasticity of functional synapse clusters such as those observed in vivo requires local rather than global mechanisms, and (4) in turn, local plasticity by small synaptic clusters may lead to powerful tuning of somatic responses.

## Methods

**Simplified model for structural plasticity**. The simplified model neuron was composed of 10 dendritic and 1 somatic subunit. The activity of dendritic subunit $j$ was the nonlinear function of the sum of its inputs:

$$a_j = f\left(\sum_{i \in S_j} s_i\right) \quad (2)$$

where $S_j$ contains the indices of the presynaptic inputs targeting subunit $j$ and $s_i$ is the spike count at input $i$. We used a sigmoid subunit nonlinearity with parameters $\zeta$ and $\theta$ controlling its slope and the threshold, respectively:

$$f(x) = \frac{1}{1 + \exp(-\zeta(x - \theta))} \quad (3)$$

The activity of the somatic subunit was the sum of the dendritic activations, and we used a fixed somatic spiking threshold $\theta_{sp}$ to calculate the binary output of the cell, $r(t)$.

The inputs were organised into 10 different ensembles each fluctuating between a low ($\lambda = 1$ Hz) and a high ($\lambda = 10$ Hz) activity state with only one of the ensemble active at any given time, and the activation of a particular ensemble corresponded to a specific spatial location. Spike counts were generated from a Poisson process in 100 ms time bins, corresponding to theta cycles. Inputs were either selected uniformly (10 from each ensemble, Fig. 1c) or randomly (Fig. 1d, e) from the 10 ensembles.

To model structural plasticity, we used a postsynaptically gated plasticity rule where the stabilisation of the synapse was controlled by synapse-specific factor $0 \leq \phi_i(t) \leq 100$, influenced by the spike count $s_i$ of input $i$ and the output of the cell $r(t)$ (Eq. (1)) with parameters $\alpha = 5$ and $\beta = 1$. We initialised $\phi(0) = \phi_{init} = 10$ for all synapses and the synapse was replaced by an other input selected randomly from the presynaptic ensembles if $\phi_i(t)$ became ≤0, when $\phi_i(t)$ was also reset to $\phi_{init}$.

We simulated plasticity with near-linear ($\theta = 5$, $\zeta = 0.35$, $\theta_{sp} = 3.3$) and superlinear ($\theta = 6.5$, $\zeta = 20$, $\theta_{sp} = 0.95$) subunit activation functions in 25 independent neurons for 10,000 time bins and analysed the connectivity between presynaptic ensembles and the subunits. The randomness of the cluster-size distribution was assessed by first creating 100 mean-matched controls by randomly shuffling the synapse identities between subunits in each time step. We then compared the observed ensemble-size distribution with the shuffled data using Pearson's Chi-squared Test. Datapoints in Fig. 1d show the $P$-value of this test for every 1 s (10 time step). The moving average (solid lines in Fig. 1d) was calculated from the log of the $P$-value using a Gaussian filter with $\sigma = 0.3$ s.

**Table 1 Ion channel densities in the CA1 model.**

| | Na$^+$ (S cm$^{-2}$) | K$_{DR}^+$ (S cm$^{-2}$) | K$_A^+$ (S cm$^{-2}$) |
|---|---|---|---|
| Apical trunk (<100 μm) | 0.04 | 0.04 | min{0.048 (1 + $d$/100), 0.288} |
| Apical trunk (>100 μm) | min{0.04 (1 + $d$/1000), 0.06} | 0.04 | min{0.048 (1 + $d$/100), 0.288} |
| Other dendrites | 0.03 | 0.02 | 0.02 |
| Soma | 0.2 | 0.04 | 0.02 |
| Axon hillock | 0.04 | 0.04 | 0 |
| Axon initial segment | 0.04 | 0.04 | 0.02 |
| Axonal inter-nodal segment | 0.04 | 0.04 | 0.004 |
| Axonal nodes | 15 | 0.04 | 0.004 |

The distance along the trunk, $d$, is measured in μm. As in Jarsky et al.[25], the dendritic segments closer than 100 μm to the soma contained K$_A^+$ channels with lower half-inactivation voltage[27].

**Biophysical models**. All simulations were performed with the NEURON simulation environment (version 7.4) embedded in Python (version 2.7).

CA1 neuron model: We used a modified version of the CA1 pyramidal cell model of Jarsky et al.[25] to better account for the dendritic processing of synaptic inputs in CA1 pyramidal neurons [8]. The passive parameters of the model were slightly adjusted to capture the dendro-somatic attenuation of synaptic responses: $C_m = 1$ μF cm$^{-2}$, $R_i = 100$ Ωcm and $R_m = 20$ kΩcm$^2$ in the dendrites, $R_m = 40$ kΩcm$^2$ in the soma and in the axon and $R_m = 50$ Ωcm$^2$ in the axonal nodes. The excitatory synapses were placed on dendritic spines consisting of a spine neck (length: 1.58 μm, diameter: 0.077 μm) and spine head (length: 0.5 μm, diameter: 0.5 μm) with total neck resistance $R_{neck} \approx 500$ MΩ (ref. [67]). Since only about 10% of the spines present in CA1 pyramidal cells were modelled explicitly in our simulations, the effect of the remaining spines was taken into account by increasing $C_m$ and decreasing $R_m$ by a factor of 2 in dendritic compartments beyond 100 μm from the soma.

To replicate important features of dendritic integration of excitatory synaptic inputs in the model we had to slightly modify the original ion channel parameters. We focused on local nonlinearities (i.e., Na$^+$ and NMDA spikes) as we assumed that such spikes are more likely to be engaged by small synaptic clusters than global regenerative dendritic spikes such as plateau potentials[68]. Specifically, in order to increase the threshold for dendritic Na$^+$ spike initiation to a similar value as the threshold for local NMDA spikes[8], the activation curve of the Na$^+$ channels was shifted by 20 mV in the basal, oblique and tuft branches (but not in the apical trunk). Furthermore, to prevent the attenuation of NMDAR currents by the activation of K$^+$ channels after local Na$^+$ spikes, we decreased the density of the K$^+$ channels in the dendritic branches (Table 1). Finally, in the original model[25] action potentials were initialised in the axonal nodes and propagated actively to the soma. Under in vivo-like synaptic inputs, the high conductance load at the soma often prevented the generation of full action potentials in the original model and in these cases axonal spikes appeared as spikelets. To eliminate spikelets we increased the somatic and decreased the axonal Na$^+$ channel conductance (Table 1).

For the simulations with strongly excitable dendritic branches (Fig. 5, Supplementary Figs. 5c–e and 6) we selected 12 terminal branches longer than 60 μm, and added a hotspot of voltage gated Na$^+$ channels[69] by increasing the sodium conductance to 0.3 S cm$^{-2}$ in the middle 30% of those branches.

L2/3 neuron model: For the L2/3 pyramidal neuron model shown in Supplementary Fig. 4 we used a detailed reconstruction of a biocytin-filled layer 2/3 pyramidal neuron (NeuroMorpho.org ID Martin, NMO-00904) as described previously[13,43]. The passive parameters were $C_m = 1$ μF cm$^{-2}$, $R_m = 7000$ Ωcm$^2$, $R_i = 100$ Ωcm, yielding a somatic input resistance of 70 MΩ.

Active conductances were added to all dendritic compartments and to the soma and included the following: voltage-activated Na$^+$ channels (soma: 100 mS cm$^{-2}$, dendrite: 8 mS cm$^{-2}$ and hotspots[69]: 60 mS cm$^{-2}$); voltage-activated K$^+$ channels (10 mS cm$^{-2}$ soma and 0.3 mS cm$^{-2}$ dendrite); M-type K$^+$ channels (soma: 0.22 mS cm$^{-2}$ and dendrite: 0.1 mS cm$^{-2}$); Ca$^{2+}$-activated K$^+$ channels (soma: 0.3 mS cm$^{-2}$ and dendrite: 0.3 mS cm$^{-2}$); high-voltage activated Ca$^{2+}$ channels (soma: 0.05 mS cm$^{-2}$ and dendrite: 0.05 mS cm$^{-2}$) and low-voltage activated Ca$^{2+}$ channels (soma: 0.3 mS cm$^{-2}$ and dendrite: 0.15 mS cm$^{-2}$). Calcium handling was modelled by a first-order system representing Ca$^{2+}$ pumps and buffers with a time constant of decay of Ca$^{2+}$ of $\tau = 28.6$ ms and the equilibrium free intracellular Ca$^{2+}$ concentration of $C_{Ca} = 100$ nM.

Synapses: The model included AMPA and NMDA excitation and slow and fast GABAergic inhibition with synaptic conductances modelled as double-exponential functions. Each excitatory synapse included an AMPA and a NMDA component which were thus colocalized and always coactivated. Similarly, inhibitory synapses were composed of a mixture of GABA-A and GABA-B receptors. The parameters of the synaptic conductances are shown in Table 2 for both the L2/3 and the CA1 neuron model. The voltage dependence of the NMDAR conductance was captured by the standard sigmoidal activation curve:

$$g_{NMDA} = \bar{g}_{NMDA} \left(1 + \frac{C_{Mg}}{4.3} e^{-0.071V}\right)^{-1} \qquad (4)$$

with the Mg$^{2+}$ concentration beeing $C_{Mg} = 1$ mM and with a slightly steeper

**Table 2 Synaptic parameters used in the models.**

| | | CA1 model | L2/3 model |
|---|---|---|---|
| AMPA | $\tau_1$ | 0.1 ms | 0.1 ms |
| | $\tau_2$ | 1 ms | 2 ms |
| | $g_{max}$ | 0.6 nS* | 0.5 nS |
| | $E_{rev}$ | 0 mV | 0 mV |
| NMDA | $\tau_1$ | 2 ms | 2 ms |
| | $\tau_2$ | 50 ms | 40 ms |
| | $g_{max}$ | 0.8 nS* | 0.8 nS |
| | $E_{rev}$ | 0 mV | 0 mV |
| GABA$_{fast}$ | $\tau_1$ | 0.1 ms | 0.1 ms |
| | $\tau_2$ | 4 ms | 4 ms |
| | $g_{max}$ | 0.7 nS | 0.7 nS |
| | $E_{rev}$ | −65 mV | −70 mV |
| GABA$_{slow}$ | $\tau_1$ | 1 ms | 1 ms |
| | $\tau_2$ | 40 ms | 40 ms |
| | $g_{max}$ | 1.2 nS | 0.33 nS |
| | $E_{rev}$ | −80 mV | −85 mV |

*The maximal conductance of the AMPA (NMDA) synapses was chosen randomly from a uniform distribution between 0.15 (0.2) and 1.05 (1.4) nS for the CA1 model during random inputc conditions, respectively.

activation than in the original model of Jahr and Stevens[70]. The maximal conductance of both the NMDA and AMPA synapses was doubled for the clustered synapses when we modelled LTP.

We simulated $N_E = 2000$ excitatory and $N_I = 200$ inhibitory synapses in the CA1 cell and $N_E = 1920$ excitatory and $N_I = 192$ inhibitory synapses in the L2/3 cell. During random connectivity excitatory synapses were placed randomly with a uniform distribution on the entire dendritic tree of the postsynaptic neuron (Fig. 2f, Supplementary Fig. 3a).

When we systematically varied the size of the clusters, we selected 240 presynaptic inputs based on the location of their place field (or orientation tuning in L2/3 cell) and varied the configuration of the corresponding synapses on the postsynaptic dendritic tree. Specifically, the inputs were ordered by the location of their place fields (orientation preferences) and were divided into $N_{clust} \in$ {240, 120, 48, 24, 12, 8, 4} contiguous and disjoint sets of clusters, each containing $M_{clust} \in$ {1, 2, 5, 10, 20, 30, 60} inputs. The remaining 1760 background inputs, having tuning curves negatively correlated with the tuning of the clusters, were placed randomly with uniform density. Clusters were placed on dendritic branches longer than $L = 60$ μm by first randomly selecting a branch (with probability proportional to its length) and then a cluster starting point between the proximal end of the branch and $L - d\,M_{clust}$ μm distance from it. Synapses were added distally from the cluster starting point with inter-synapse distance of $d = 1$ μm. This procedure guaranteed that synapses within one cluster target the same dendritic branch, show maximally correlated activity and that the expected location of clustered synapses is independent of the cluster size. Note, that selection of long branches, typically thin terminal dendrites, for the location of clustered synapses introduces a slight bias for the postsynaptic processing of clustered inputs even when $M_{clust} = 1$.

For the simulations shown with open symbols in Fig. 3e, f the 20 synapses forming each of the 12 clusters were randomly redistributed along the entire dendritic branch receiving clustered inputs.

Dendritic branches receiving clustered inputs had higher input rate due to the presence of additional, background inputs. This was not a problem during theta as the background activity is relatively low and asynchron in that case, but could be a substantial concern during sharp waves. To exclude the possibility that synaptic

saturation during SPWs is caused by the increased input density at clustered branches, we generated 2000 (instead of 1760) background input locations distributed uniformly and selected the $M_{clust}$ background inputs closest to the location of the clustered synapses. These background inputs did not participate in the replay of any particular trajectory but only fired at elevated baseline rate during SPWs (see the section Inputs below). Conversely, the clustered synapses did not show an elevated baseline activity during SPWs but only participated in the trajectory replay. Using this procedure we achieved a similar average input level for branches receiving clustered vs. non-clustered inputs. The data presented in this paper used this more uniform input distribution during SPWs, but similar results were achieved when clusters were simply added to the background (data not shown).

Inhibitory synapses were divided into two groups with 80 synapses targeting the soma (and the apical trunk in the case of the CA1 neuron) and the remaining synapses were distributed randomly along the entire dendritic tree.

Balanced connectivity: To achieve a fully balanced synaptic connectivity (denoted as $F$ in Fig. 2) in the case of the hippocampal pyramidal cell, we first applied co-prime reordering procedure on the $N = 40$ presynaptic ensembles (see below) that minimised correlation between the neighbouring ensembles. To achieve this, we selected base numbers $\alpha$ relative prime to $N$ and generated an arithmetic progression with difference $\alpha$, dividing the elements of the sequence by $N$ and taking the remainder (Fig. 2f). In the case of $\alpha = 3$ and $N = 10$ a potential sequence is:

$$\{2, 5, 8, 11, 14, 17, 20, 23, 26, 29\} \bmod 10 = \{2, 5, 8, 1, 4, 7, 0, 3, 6, 9\}$$

In this sequence the distance between neighbouring elements is constant, so the similarity between any pair of neighbouring input is identical and thus input correlations are the same within any contiguous subset of the sequence. Also note that the sequence is not repeating until $N$ elements. We chose $\alpha = 9$ that minimised correlation between inputs within ~40 μm (9 synapses).

Next, we arranged the 2000 inputs into a single sequence where cells from the 4 different different ensembles were selected according to the ordering defined above. Finally, we mapped this sequence to the dendritic tree of the neuron starting from the soma and following each subtree towards the distal end sequentially with constant distance (d=4.877 μm) between the synapses. This defined the fully balanced (F) connectivity (Supplementary Fig. 3d).

In the locally balanced connectivity (L in Fig. 2) we started with a random connectivity and applied the co-prime reordering procedure independently to individual dendritic branches. Specifically, we ordered the $N_i$ synapses targeting branch $i$ according to their place field location, selected a co-prime $\alpha_i \approx N_i/4$ and rearranged the inputs within branch $i$ while also equalising the distance between neighbouring synapses (Supplementary Fig. 3b).

In addition, we also tested a globally balanced connectivity (G in Fig. 2), where we started from the fully balanced connectivity, and randomised synapse location and the ordering of the synapses within each branch separately (Supplementary Fig. 3c).

Visual cortical inputs targeting the L2/3 neuron model had 4 different features: orientation preference, phase preference, orientation selectivity and response linearity[71]. We co-sorted inputs to achieve an input distribution that is approximately balanced with respect to all 4 features at the same time (for details, see legend of Supplementary Fig. 3e–h).

**Inputs.** To study the impact of synaptic clustering under in vivo-like conditions in the biophysical models we carefully matched the inputs of the models to the synaptic inputs experienced by these neurons in vivo. Inputs, corresponding to synaptic release events, were generated by a Generalised Linear Model (GLM) tuned to a number of different external and internal features[72,73]. Specifically, the presynaptic input count at time $t$ for synapse $i$ were generated from a Poisson process with rate $\lambda_t^i$:

$$s_t^i \sim \text{Poisson}(\Delta t\, \lambda_t^i) \tag{5}$$

The firing rate $\lambda_t^i$ of the presynaptic neuron $i$ at time $t$ was defined as

$$\lambda_t^i = \exp(a_t^i) + \lambda_0^i \tag{6}$$

$$a_t^i = \mathbf{w}_\phi^i\, \boldsymbol{\phi}(x_t) + \mathbf{w}_\psi^i\, \boldsymbol{\psi}(t) + \mathbf{w}_\chi^i\, \boldsymbol{\chi}(\mathbf{s}^i) \tag{7}$$

where $a_t^i$ is called the activation and $\boldsymbol{\phi}(x_t)$, $\boldsymbol{\psi}(t)$ and $\boldsymbol{\chi}(\mathbf{s}^i)$ denote the activation of basis functions tuned to external inputs (location or stimulus orientation and phase), internal processes (phase of theta or gamma oscillation) or the output of presynaptic neuron $i$, respectively. The tuning properties of a particular input is set by the nature of these basis functions, varied among different conditions (theta or SPW for CA1 and gratings for L2/3) and the parameter vector $\mathbf{w}^i$ specific for each cell $i$. The term $\mathbf{w}_\chi^i\, \boldsymbol{\chi}(\mathbf{s}^i)$ captures the effect of past events on the input rate[72] and was included to model a short (~5 ms) refractory period in the inputs.

Although the rate of each input was a deterministic function of time, the synaptic events were generated by a stochastic process accounting for variability in spike timing and stochastic vesicle release. This allowed us to generate multiple trials with identical input rates to estimate the trial-to-trial variability. With both

the balanced and the random connectivity we generated 10 different synaptic arrangements and input populations and simulated 16 trials with each of them.

Although the framework is identical for the different conditions, we describe our choice for the basis functions and the parameters separately in the following sections. Inputs were generated using custom written programmes in R.

Hippocampal inputs during theta: To simulate the inputs to the CA1 pyramidal cell we focused on the population activity of the presynaptic CA3 pyramidal neurons and ignored inputs arriving from elsewhere, including the entorhinal cortex. A single CA1 pyramidal neuron receives $N_{syn} \approx 20{,}000$ excitatory synapses[74], and ≈10% of the presynaptic CA3 cells have place field in a typical environment[29,30]. Therefore we simulated $N_{PC} = 2000$ place cells each exhibiting a single place field in the environment, while the activity of the other 18,000 cells was incorporated in the increased baseline firing rate of the simulated place cells.

In the simulations the animal was moving at a constant 20 cm/s speed on a 2 m-long circular track. Excitatory neurons had a single, idealised place field that showed phase precession relative to the ongoing theta oscillation (constant 8 Hz frequency). Phase precession was modelled using basis functions co-tuned to spatial location and theta phase. Specifically, we had $N_{basis} = N_x \cdot N_\psi = 160$ Gaussian basis functions with $N_x = 40$ spatial and $N_\psi = 4$ temporal components uniformly tiling the space with standard deviation $\sigma_x = 5$ cm and $\sigma_\psi = \pi/2$ radians. The parameter $\mathbf{w}$ was optimised numerically to match the phase precession obtained from Skaggs et al.[28] with the diameter of the place field being $d \approx 30$ cm (Supplementary Fig. 1i). To tile the space evenly, we shifted the parameter $\mathbf{w}_x$ along the spatial dimension either randomly (Fig. 2b–e; 10 presynaptic populations of 2000 place cells with random place field properties) or uniformly (Fig. 2g–i, 40 ensembles and 50 cells with identical tuning in each ensemble). The average firing rate of the presynaptic place cells was either chosen randomly from a gamma distribution[37] with shape and rate parameters $\alpha = 6$ and $\beta = 6$ (random) or was identical 1 Hz for all neurons (uniform) corresponding to a 5 Hz presynaptic firing rate combined with the low release probability of hippocampal synapses[75] ($p_{rel} \approx 0.2$). To model spatially untuned activity of non-active place cells the parameter $\lambda_0$ was set to 1 Hz leading to the average input rate $\hat{r}_{PC} = 2$ Hz for the simulated place cells. The total event rate of the excitatory inputs was thus $r_{tot} = N_{PC} \cdot \hat{r}_{PC} = 4$ kHz matching the event rate a pyramidal neuron is expected to receive in vivo, $r_{tot} = N_{syn} \cdot \hat{r} \cdot p_{rel} \approx 4$ kHz, where $\hat{r} \approx 1$ Hz is the average firing rate of a randomly chosen hippocampal pyramidal neuron[76].

Hippocampal inputs during SPW: To simulate SPW events we embedded elevated hippocampal population activity of $T_{SPW} = 100$ ms duration in a low activity baseline state with independent presynaptic activity at a constant 0.8 Hz event rate. During the SPW the inputs were driven by a simulated spatial trajectory corresponding to memory replay[36]. The presynaptic cells had a spatial tuning and were also modulated by ongoing 150 Hz ripple oscillation[35]. Specifically, we used $N_{basis} = N_x + N_\psi = 44$ Gaussian basis functions with independent spatial ($N_x = 40$) and temporal ($N_\psi = 4$) components. The place fields had a diameter of $d \approx 50$ cm and were distributed uniformly across the entire track and the peak firing rate of the place cells was identical.

The average number of spikes fired by a rodent hippocampal pyramidal cell during an individual sharp wave event of $T_{SPW} = 100$ ms duration was $N_{sp} \approx 0.4$ (ref. [37]), leading to a $N_{syn} \cdot N_{sp} \cdot p_{rel} = 1600$ synaptic events received by the postsynaptic cell. During the SPW a part of the previously experienced trajectory of the animal is replayed with an increased speed of $v \approx 6$ m s$^{-1}$ (ref. [38]). In our 2-m long environment ≈30% of the track is replayed in each SPW by the $N_{replay} \approx 600$ neurons having place fields overlapping with the replayed trajectory. We assume that these place cells are firing at an increased mean rate of $r_{replay} \approx 60$ Hz and are thus responsible for $N_{replay} \cdot r_{replay} \cdot T_{SPW} \cdot p_{rel} = 720$ synaptic events, while the remaining 880 inputs are uniformly distributed across the entire presynaptic population (19,400 cells, out of which we simulated only 2000) by increasing the baseline firing rate of the place cells uniformly during the SPW period.

For the 240 inputs participating in synaptic clusters, the constant baseline firing was not increased, as these inputs represented input from a single presynaptic neuron. Instead, we selected the 240 of the 2000 not clustered synapses with location closest to the location of the clustered inputs, and these synapses received inputs with only elevated baseline activity during SPW but showed no spatial tuning.

The firing rate of the inhibitory inputs targeting the perisomatic (dendritic) region switched from $r_{bg}^{basket} = 10$ Hz ($r_{bg}^{dend} = 5$ Hz) baseline rate to $r_{SPW}^{basket} = 30$ Hz ($r_{bg}^{dend} = 15$ Hz) during SPWs, respectively. Both excitatory and inhibitory inputs were modulated by the 150 Hz ripple oscillation with the ratio between their peak/minimal firing rate being $r_{peak}/r_{min} \approx 3$ (ref. [31]). The depolarisation amplitude and the ripple modulation of the somatic membrane potential during SPWs in the biophysical model was consistent with data recorded from awake mice[39,77].

We varied the starting position of the replayed trajectory in steps of 10 cm changing the overlap between the represented trajectory and the place field of the clustered neurons and analysed the average somatic response during SPWs in 16 independent trials.

L2/3 inputs: To generate in vivo-like synaptic inputs to a L2/3 pyramidal cell we simulated the activity of cortical and thalamic neurons in response to drifting grating stimuli, widely used to study neuronal coding in the visual system. Since the

activity of these two populations is reasonably similar under these conditions[78], we did not treat them separately.

We modelled the response of visual neurons to gratings moving at 2 cycles/s with 16 different directions each shown for 1.5 s. The activity of each input was tuned to the direction and the phase of the stimulus with the distribution of firing rates, orientation selectivity, direction selectivity, response linearity (phase modulation depth divided by the mean firing rate; F1/F0 calculated as in Niell and Stryker[71]) and the width of the tuning curve matched to experimental data[71,78] (Supplementary Fig. 4b). We used $N_{basis} = N_\theta + N_\varphi + N_\psi = 28$ circular Gaussian basis functions with independent components responsible for the tuning to motion direction ($N_\theta = 16$), phase ($N_\varphi = 6$) and gamma oscillation ($N_\psi = 6$). Cells were divided into 16 groups based on their direction preference, and within each group we simulated 120 neurons with 5 different directional tuning selectivity (including direction-selective and orientation-selective cells), 6 different preferred stimulus phase and 4 different phase tuning selectivity, including simple cells (showing marked phase preference) and complex cells (having little phase preference). In the random condition (Supplementary Fig. 4d, e, random inputs) the orientation and stimulus phase preference of the cells were selected randomly with uniform probability and their average firing rate was chosen randomly from a gamma distribution with shape and rate parameters $\alpha = 2 \cdot 4.17$ and $\beta = 2$. In the uniform condition (Supplementary Fig. 4, except panels d, e, uniform inputs, f–j), each presynaptic input represented a unique combination of these features with identical mean firing rate $\hat{r}_{LGN} \approx 4.17$ Hz. The total excitatory input rate of the L2/3 model neuron was around 8 kHz.

Inhibitory inputs had weak orientation tuning, showed complex cell-like phase preference and had a mean firing rate of $r_{inh} \approx 30$ Hz. We generated 10 different presynaptic populations with either uniform or random tuning parameters and simulated 16 trials with each population.

**Decomposition of response variance.** The subthreshold membrane potential response $r(x, k)$ of the postsynaptic neuron at location $x$ in trial $k$ can be approximated by the sum of stimulus-dependent (s) and dendritic (d) factors, each factor potentially contributing both to the tuning curve ($\mu$) and to trial-to-trial variability ($\Phi$) of the responses:

$$r(x, k) = \mu_s(x) + \Phi_s(x, k) + \mu_d(x) + \Phi_d(x, k) \quad (8)$$

where $\mu_i(x)$ captures the systematic, location dependent changes in the mean response and $\Phi_i(x, k)$ is the drive fluctuating from trial-to-trial that is associated with the given factor. Specifically, $\Phi_s$ is associated with fluctuation in the total input (presynaptic spike counts) arriving to the cell while $\Phi_d$ denote fluctuations in the input related to the location of the activated synapses within the dendritic tree brought about by passive (e.g., cable filtering) and active (e.g., location dependent activation of NMDA receptors) mechanisms, as well as functional synaptic clustering (nonlinear interactions between neighbouring active inputs). Importantly, we assume that the means of the trial-to-trial fluctuations are zero and their variances are constant:

$$E_{x,k}[\Phi_s] = 0$$
$$Var_{x,k}[\Phi_s] = \varsigma_s^2$$

with similar equations for $\Phi_d$. Note, that our derivation can be generalised by defining $x$ as a point in a low dimensional manifold of the presynaptic firing rates.

We defined these factors to have a linear effect on the somatic response (membrane potential), so biophysically these factors correspond to currents flowing into the soma of the neuron. Note, that these factors do not correspond to separate biophysical processes: during the sustained synaptic background activity, the activation of a single additional synapse contributes to both dendritic and input factors as it increases the total input count (input factor) but the local depolarisation it causes also modulates input current flowing though the neighbouring synapses (dendritic factor). Also note, that in general, these factors are not independent of each other, as larger input strength ($\Phi_s$) is potentially associated with larger NMDAR current or stronger interactions within the clusters ($\Phi_d$).

In contrast to our simplified model used for studying plasticity, these factors can not be directly measured in real neurons or in a biophysical model. Therefore, instead of pinpointing the effect of these factors on a trial by trial bases, we were aiming at identifying their average contribution to the neuronal response variability. To achieve this, we derived a novel analysis technique, the decomposition of neuronal response variance where we selectively modified the contribution of the individual factors and measured their effect on the response in a biophysical model.

To estimate the contribution of the specific factors, we performed 4 different types of simulations. In the first scenario we used random inputs and random connectivity and thus all factors were present and the sum of their contribution to the response variance was measured (random case).

In the second scenario, which we called the uniform input scenario $\mu_s(x) = \mu_s$, so the mean total input was independent of the position, and all fluctuations in the response were caused by either trial-to-trial variability or systematic changes in the dendritic factors. Moreover, we assume that the dendritic factors were identical in the uniform and in the random case, that is, we assumed that changing the input heterogeneity did not change the contribution of dendritic factors. This is a critical

assumption that is not justified in most cases (e.g., when large, local fluctuation in the inputs can boost dendritic nonlinearities), but can be a reasonably good approximation when changes in the input strength are relatively small. Therefore we used this analysis only when the connectivity of the synapses was random or balanced (i.e., not clustered), and therefore we did not expect large, systematic fluctuations in the synaptic drive on any given dendritic branch. In the uniform scenario Eq. (8) reads as:

$$r_v(x, k) = \mu_s + \Phi_s(x, k) + \mu_d(x) + \Phi_d(x, k)$$

where the lower index in $r_v$ indicates the condition ($v =$ "uniform").

In the third simulation type, named as fully balanced we removed the dendritic factors by arranging synapses regularly throughout the entire dendritic tree. Instead of entirely eliminating the effect of dendritic processing, we only equalised its effect on the differently tuned inputs, so that dendritic components did not add to the variability of the neuronal tuning. This way we could selectively manipulate the effect of dendritic factors on the somatic response, without changing the contribution of the input factors. In the fully balanced scenario the systematic variability in the tuning was caused by input components, and we assumed that the contribution of the input components were identical in the fully balanced and in the random case. In the fully balanced scenario Eq. (8) simplifies to:

$$r_\beta(x, k) = \mu_s(x) + \Phi_s(x, k) + \mu_d + \Phi_d(x, k)$$

where the lower index in $r_\beta$ indicates the scenario ($\beta =$ "balanced").

As a control, we performed a fourth simulation type, where both the dendritic and input factors were eliminated. In this case we expected the postsynaptic tuning to be flat, as all variability was caused by trial-to-trial fluctuations:

$$r_{v\beta}(x, k) = \mu_s + \Phi_s(x, k) + \mu_d + \Phi_d(x, k)$$

We denote the trial-to-trial variance of one component of the drive with $\varsigma_i^2$, with $i = \{\beta, v\}$ coding for trial type, and assume that they do not change between the different scenarios. The correlation between the trial-to-trial components effect of the somatic and dendritic factors is denoted by $\rho_{sd}$. The trial-to-trial variance of the response is expected to be identical in all scenarios:

$$\varsigma^2 = \varsigma_s^2 + \varsigma_d^2 + 2\varsigma_s\varsigma_d\rho_{sd} \quad (9)$$

where $\varsigma^2$ can be measured in a biophysical model. Confirming our assumption that changing the input and the connectivity does not change the trial-to-trial variability, we found that $\varsigma^2$ was similar in all four scenarios in the biophysical model of both the hippocampal (Fig. 2k) and the visual cortical (Supplementary Fig. 4e) neuron.

When we average over $N$ trials, the variance of the mean response (the estimated tuning curve) at a particular location will be $\varsigma^2/N$. Denoting the variance of the location dependent response by $\sigma_i^2 \triangleq Var[\mu_i(x)]$, the variance of the estimated tuning curve in the four scenarios can be written as:

$$\sigma^2 = \varsigma^2/N + \sigma_s^2 + \sigma_d^2 + 2\sigma_s\sigma_d\rho_{sd}$$
$$\sigma_v^2 = \varsigma^2/N + \sigma_d^2$$
$$\sigma_\beta^2 = \varsigma^2/N + \sigma_s^2$$
$$\sigma_{v\beta}^2 = \varsigma^2/N$$

We identified the contribution of stimulus-dependent and dendritic factors with the variances $\sigma_s^2$ and $\sigma_d^2$ (Fig. 2l), which we calculated from the tuning curve variances ($\sigma^2$, $\sigma_v^2$, $\sigma_\beta^2$ and $\sigma_{v\beta}^2$; Fig. 2k) measured in the four different scenarios.

**Data analysis.** To calculate the tuning curve from the recorded somatic membrane potentials we first detected action potentials (AP) as positive crossing of a threshold of $V_{th} = -30$ mV. We obtained the subthreshold response by replacing the APs by the voltage before the start of the AP, defined as the first point when the derivative of the voltage exceeded $\eta = 5$ mV ms$^{-1}$ before the spike. The end of the AP was defined as the time when the membrane potential fell below threshold.

Next, we filtered the raw subthreshold response with a Gaussian kernel which removed the oscillatory component from the sVm yielding the slow $V_m$ response. The width (SD) of the Gaussian kernel was $\sigma_{theta} = 100$ ms in the CA1 cell during theta, $\sigma_{ripple} = 2.5$ ms during SPWs and $\sigma_{gamma} = 100$ ms in the case of the L2/3 cell. We applied a similar Gaussian filtering with $\sigma_{theta} = 100$ ms to the input spike counts when we evaluated the input variability (Fig. 2b, g, j and Supplementary Fig. 4d).

Finally the tuning curve was obtained by averaging the slow $V_m$ responses across 16 trials with identical presynaptic firing rates but random synaptic events. The variability of the tuning curve was calculated as the variance of the points of the tuning curve along the track. The trial-to-trial variability was calculated by first computing the variance of the 16 trials and then taking the average along the track. The response integral during theta stimulation was calculated by first subtracting the baseline $V_m$ defined as the mean sVm in the last 2000 ms from the individual slow $V_m$ responses, and then integrating the baseline shifted response over time. The SPW amplitude (Fig. 4b, e) was defined as the difference between the maximum of the average filtered somatic membrane potential ($V_{resp}$) during the SPW and the minimum of $V_{resp}$ before the SPW. The excess response amplitude (Fig. 4h) was defined relative to the one synapse per cluster response as the

difference in the SPW amplitude or as the difference in the mean depolarisation within the place field (dashed line in Fig. 3c).

To estimate the frequency of dendritic spikes we recorded the membrane potential of 4 dendrites selected randomly among the branches receiving clustered input. We defined the dspike ratio as the proportion of time the local $V_m$ was above the dendritic spike threshold during the activation of the clustered synapses (dashed grey lines in Figs. 3a and 4a). Since the threshold for generating sodium and NMDA nonlinearities approximately coincides both in CA1 neurons[8] and in our model, we used the membrane potential with the maximal NMDA current ($V_m = -25$ mV; Fig. 4f) as dendritic spike threshold.

**Reporting summary**. Further information on research design is available in the Nature Research Reporting Summary linked to this article.

## Data availability

The source data underlying Figs. 2j, k, 3e–g, 4d, e, g, h and 5c, d are provided as a Source Data file. The datasets generated and analysed in the current study can be reproduced using the computer codes provided.

## Code availability

The code used for simulating the biophysical model and generating the inputs can be downloaded from the online repositories https://bitbucket.org/bbu20/clustering and https://bitbucket.org/bbu20/popact, respectively.

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

## Acknowledgements

This work was supported by an NKFIH fellowship (PD-125386, FK-125324; B.B.U.), the International Research Scholar program of the Howard Hughes Medical Institute (55008740; J.K.M.), and the European Research Council under the European Union's Horizon 2020 research and innovation program (CoG 771849; J.K.M.). We thank Lajos Vágó for the idea of co-prime ordering, Zoltán Nusser for discussions and for his comments on an earlier version of the manuscript and one of our anonymous Reviewers for encouraging us to extend our investigations to the SPW state.

## Author contributions

B.B.U. and J.K.M. designed the study. B.B.U. performed the simulations and analysed the data. J.K.M. and B.B.U. interpreted the results and wrote the paper.

## Competing interests

The authors declare no competing interests.
