## [Peer Review File · Nature Communications]

Reviewers' Comments:

Reviewer #1:

Remarks to the Author:

Balazs et al. describe a novel model and computational framework to examine how the dendritic organization of synaptic inputs and dendritic nonlinearities contribute to the formation of sensory receptive field properties. Their model involves the construction of realistic biophysical neurons (hippocampus CA1, L2/3 visual cortex) with realistic input dynamics. Their work has several findings: (1) global activity can indeed potentiate random groups of co-active synapses to form functional clusters, (2) clusters containing 10-20 co-active synapses lead to reliable formation of receptive fields and corresponding somatic membrane potential responses, and (3) global events such as sharp wave ripples in the hippocampus reduce the effect of co-active inputs. This modeling approach is impressive and novel in its own right. It will be of value to researchers in a number of fields--for both in vitro and in vivo neurophysiologists. This model will undoubtedly be refined in the future as new experiments provide insights into the properties of individual neurons, but it provides an important foundation to build upon.

Major concerns:

- The authors pose an idea in the beginning of the paper based on real data (co-active inputs are part of distributed small clusters), and contrast in vivo data with in vitro experiments. In the end, the model appears to support in vitro but not in vivo observations. What are the possible reasons for this? Are there other factors that could be contributing or limit the size of synaptic clusters?

-Did the authors examine the number of clusters per dendrite? Can this play a role or reduce the total size of a given cluster on a dendritic branch?

-Is it necessary to discuss both cell types? Because most everything in the paper is focused on CA1, and the visual cortex analysis is limited to a single supplementary figure, it seems more appropriate to mention briefly in the Results or Discussion that the analyses were also applied to the visual cortex.

-It is not clear why it is necessary to begin the paper with a simple simulation of global plasticity. In fact, given the brevity of text and the fact that the associated figure is in supplemental, it seems like an afterthought. It would be helpful to reframe the paper to include this section more prominently and describe it more thoroughly or, perhaps, move it to the end of the paper. Relatedly, measurement of variance needs to be explained clearly in this section. Currently there is no description until later in the Results and the reader has to search through the Fig S1 legend to find it.

-There are some challenges in relating the methods on decomposition of response variance to Fig 1 plots. Please make this clear and have a clear association between symbols and experiment types. For example, the letters in Fig1J describing the arrangements need to be made clear in the text. Also, there is little discussion of Figure 1K either in the Results or Methods.

-Another major point that is not discussed is inhibition. Although little is known about individual inhibitory inputs and their organization, in the author's framework all inhibitory inputs are assumed to be untuned. Given the evidence for tuned inhibitory neurons in the visual cortex, the authors should provide some consideration of how this impacts the conclusions that are drawn .

Minor concerns :

- Please check the colors in Fig. S5J for accuracy.

- In general, the details of figures should be made less confusing for the reader. For example in Figure

1, it is difficult to figure out what all the letters stand for.

- Fig S1F: it is unclear what the colored plots represent

- Fig S1 D-E: it is not clear what these symbols are supposed to convey.

Reviewer #2:

Remarks to the Author:

The functional impact of synaptic clustering has received a large amount of attention over the years and for good reason. The authors here continue a line of work where they examine the potential impact of dendritic nonlinear synaptic integration (that requires clustering) on the neuronal input output transformations occurring during complex patterns of input in computer models. While this is possibly a fruitful exploration the main difficulties arise from the authors' capability to accurately capture the large number of parameters in play here or in this case to even sufficiently explore the parameter spaces. This paper presents additional simulations that with the current set of parameters show, again, that local dendritic nonlinearities are too weak to heavily impact the simulated response of a single neuron to two different input patterns (theta/ place field like or SPW/ripple like). In the end, this is an empirical question and it is not clear how this manuscript aids in the design and interpretation of experiments meant to provide an answer. Specifics follow.

1) One thing that might help with the impact is for the authors to use their models to explore the parameter spaces in search of models where clustering and the nonlinear dendritic integration hypothesized to detect it would actually impact the firing of a neuron during place fields or ripples. It springs to mind that the dendritic branch excitability plasticity studied in the past by one of the authors might change the results in figure 3 substantially. What happens if input is clustered on to branches with strongly propagating local Na spikes?

2) What is the role of input clustering in synaptic plasticity? The authors focus on what they call global plasticity (repeated pairing of output and input) vs local plasticity (not dependent on output). However, neither of these forms of synaptic potentiation are actually involved in the generation of place fields. Since the potentiation that does underlie place fields requires the overlap in time and space of a locally generated synaptic signal and a large dendritic depolarization (Ca²⁺ plateau) it should be interesting to explore how clustering impacts the generation of the local NMDA dependent signals. This will of course require the implementation of dendritic spines with a high resistance necks (which should have been present in all the current simulations and is a good example of how difficult it is to keep track of all the important parameters in a model like this). One additional factor to include is the regulation of spine neck resistance to counter spine saturation following LTP (see Harnett lab).

3) One final point. It would be of interest to have the authors explain what type of information is contained within dendritic clusters in CA1. Perhaps I misread the paper but the authors seem a bit vague about the hypothetical functions of clustering and their detection by dendritic nonlinearities. The old idea has been that this additional layer of processing provided by active dendritic branches allows the neuron to increase the complexity of its feature selectivity by generating higher-order features. How do the authors see this happening in the hippocampus which is presumably already very high-order even in the CA3? In addition, CA3 neurons do not have a fixed feature selectivity (each cell can change its place field location depending on the environmental context) would this cause a problem for the downstream neurons trying to integrate this input?

Reviewer #3:

Remarks to the Author:

The paper makes several claims. One is that under conditions designed to be reflective of physiological

conditions in hippocampal CA1 pyramidal cells, small randomly occurring synaptic clusters don't influence the somatic membrane potential, in a realistic computer model. Similar results were obtained with a layer 2/3 pyramidal cell model. A second result was that larger synaptic clusters can lead to clustering-based tuning – this was known, but the key point being made here is that this is above the size of synaptic clusters which are observed in vivo. Finally, the third main result is that synaptic clustering has small impact during hippocampal sharp waves.

The first of these results is interesting, although perhaps would not completely surprising. It of course suffers from the problem that it is not possible to prove a negative. Who is to say that there is not some other cell type or physiological condition under which relatively small synaptic clusters may have a larger somatic effect? However, the modelling incorporates a lot of details which have been carefully matched to experimental data. I believe that it probably provides a good representation of what is going on in CA1, although it would be nice if this could be validated in some way. I'm not sure how much this is going to influence thinking in the field – this is what most people think anyway. That does not mean that the result should not be published.

The second result may help interpret recent experimental data – it suggests that the cluster sizes seen e.g. by Takahashi et al are too small to result in clustering based tuning. I think this is a good point.

The third result, that clusters have small impact during hippocampal sharp waves, is again negative – it is perhaps worth having in the literature the wide range of circumstances under which synaptic clusters don't much effect somatic membrane potential, but I'm not sure that is really going to have widespread impact on the neuroscience community.

Finally, I should note that the initial section (and supplemental figure S1) to some extent recapitulates previously published work by Cazé et al (Biorxiv, 2015, <https://www.biorxiv.org/content/10.1101/029330v4>). That paper explored the circumstances in which a local learning rule could lead to the formation of functional synaptic clusters, and really should have been cited by the authors. On that note, I am struggling to discover from the paper exactly *how* the cluster formation mechanism proposed here differs from that of Cazé et al, because it has been described very heuristically – I was looking for, in essence, the synaptic plasticity rule. I think that in any revision of this paper, that this should be spelled out much more explicitly. If I have understood correctly, however, the Cazé model already has local plasticity by small synaptic clusters, which I what the current authors are proposing is a better way to go than the “global plasticity” mechanism used here, and which does not seem to be well justified.

Overall, I think that this paper provides some important caveats on over-interpreting some recent experimental results – but I am not sure how much impact these caveats will have on the neuroscience community, because I'm not sure that anyone is interpreting them that way anyway. The work is convincing, as far as it pertains to the specific models investigated here – but the level of biophysical detail inherently results in lack of generality. I am not sure that this is practically a resolvable issue – perhaps what would strengthen the conclusions more would be accepting that these are investigations of a very specific pair of circumstances (CA1 and L2/3) and attempting experimental validation of these models.

Response to reviewers

We thank the reviewers for their constructive comments and support for the paper. Below we address each of their concerns (our responses are in *italic*).

Reviewer #1 (Remarks to the Author):

Balazs et al. describe a novel model and computational framework to examine how the dendritic organization of synaptic inputs and dendritic nonlinearities contribute to the formation of sensory receptive field properties. Their model involves the construction of realistic biophysical neurons (hippocampus CA1, L2/3 visual cortex) with realistic input dynamics. Their work has several findings: (1) global activity can indeed potentiate random groups of co-active synapses to form functional clusters, (2) clusters containing 10-20 co-active synapses lead to reliable formation of receptive fields and corresponding somatic membrane potential responses, and (3) global events such as sharp wave ripples in the hippocampus reduce the effect of co-active inputs. This modeling approach is impressive and novel in its own right. It will be of value to researchers in a number of fields--for both in vitro and in vivo neurophysiologists. This model will undoubtedly be refined in the future as new experiments provide insights into the properties of individual neurons, but it provides an important foundation to build upon.

We thank the reviewer for his/her positive opinion and appreciating the impact of our new approach.

Major concerns:

- The authors pose an idea in the beginning of the paper based on real data (co-active inputs are part of distributed small clusters), and contrast in vivo data with in vitro experiments. In the end, the model appears to support in vitro but not in vivo observations. What are the possible reasons for this?

Our model indicates that if the size of the functional synaptic clusters is similar to the cluster sizes typically reported in vivo, than these small synaptic clusters have very limited effect on the somatic membrane potential response of the cell. We would like to emphasize that this result does not contradict in vivo observations, as the effect of synaptic clusters on sVm has not been directly measured in vivo as far as we know. Our result suggests that the primary role of the small synapse clusters is not to influence the somatic response. We clarified this in the paper (lines 318-327):

“Although the spatial scale of the functional clusters reported in cortical neurons has been restricted to 5-10 um and 2-5 dendritic spines (Takahashi et al., 2012, Iacaruso et al., 2017, Scholl et al., 2017, Kerlin et al., 2018), current experimental techniques do not allow reliable monitoring of the activity of all synaptic inputs in a given dendritic branch and thus, these studies may underestimate the real number of synapses involved in a given synaptic cluster. Moreover, small biases in the removal of signals related to back-propagating action potential when analysing functional responses of dendritic spines can also bias the estimated correlation between nearby spines (Kerlin et al., 2018). Finally, having additional synapses with selectivity similar to tuning of a small cluster on the same branch can be equally efficient in generating clustering-based tuning as a single large synapse cluster (Fig3E-F). These considerations suggest that further improvements in the experimental techniques and analysis methods are required to estimate the size and the somatic effect of in vivo occurring synapse clusters.”

Are there other factors that could be contributing or limit the size of synaptic clusters?

There are several potential factors possibly limiting the size of the synapse clusters, ranging from specific biophysical processes mediating competition between different clusters (Kirchner et al. 2019) to the interplay of learning rules at various spatial scales (Weber et al., 2016), but their detailed analysis is beyond the scope of the present study.

-Did the authors examine the number of clusters per dendrite? Can this play a role or reduce the total size of a given cluster on a dendritic branch?

We thank the reviewer for this comment. We hypothesize that multiple similarly tuned synapse clusters targeting the same branch or having additional co-tuned synapses could function as a single larger cluster (see Losonczy & Magee, 2006). To support this idea and to test the importance of the fine scale arrangement of synapses within a dendritic branch, we selected the 20 synapse per cluster configuration, and distributed the synapses participating in the clusters randomly along the branch. We then compared the responses in the focused versus the dispersed synapses configuration, and found that the fine-scale, within-branch arrangement of the synapses did not significantly influence the responses (Fig R1). Thus, having multiple, similarly tuned small synapse clusters per dendrite or additional synapses with similar tuning to the cluster can equally contribute to clustering based tuning as single, large clusters. These simulations have been included in the results section of the revised manuscript and we discuss their implications in the Discussion (line 325).

Fig. R1. Effect of fine scale synapse arrangement on postsynaptic tuning variance (left) and response integral (right) in control (orange) and LTP (brown). The clustered data (squares) is repeated from the 20 synapse per clustered Fig. 3E-F in the manuscript. Scattering (diamonds) has a minimal impact on the response statistics.

-Is it necessary to discuss both cell types? Because most everything in the paper is focused on CA1, and the visual cortex analysis is limited to a single supplementary figure, it seems more appropriate to mention briefly in the Results or Discussion that the analyses were also applied to the visual cortex.

We think that this is indeed what we do: the L2/3 model is only briefly mentioned in the Results and the Discussion at the appropriate locations without interrupting the logic of the paper. Our primary reason to move the L2/3 model to the supplementary material was to simplify the narrative

of the paper. However, we believe that the case of the L2/3 cells is sufficiently different from the CA1 cell so that it is well worth to present the related data and emphasize the consequences.

The L2/3 neuron has a different morphology and receives input with a completely different input statistics from that of a CA1 pyramidal cell. In particular, both the baseline input firing rate and the correlations between the inputs are very different in the visual cortex and in the hippocampus, raising the possibility that the minimal cluster-size to trigger nonlinear responses could also be different. Moreover, a substantial part of the experimental data about synapse clustering has been collected in the visual cortex - therefore we thought that it is essential to perform the same analysis on visual cortical neurons. Our finding that the minimal cluster size is similar in the visual cortex and the hippocampus suggests that this might be a general principle and thereby we believe widens the impact of our study.

-It is not clear why it is necessary to begin the paper with a simple simulation of global plasticity. In fact, given the brevity of text and the fact that the associated figure is in supplemental, it seems like an afterthought. It would be helpful to reframe the paper to include this section more prominently and describe it more thoroughly or, perhaps, move it to the end of the paper. Relatedly, measurement of variance needs to be explained clearly in this section. Currently there is no description until later in the Results and the reader has to search through the Fig S1 legend to find it.

We thank the reviewer for this comment. We have substantially rewritten and expanded that section of the paper and included a simplified version of the original figure into the main text to ensure that this part of the text integrates well into the manuscript and that all the details are explained.

-There are some challenges in relating the methods on decomposition of response variance to Fig 1 plots. Please make this clear and have a clear association between symbols and experiment types. For example, the letters in Fig1J describing the arrangements need to be made clear in the text. Also, there is little discussion of Figure 1K either in the Results or Methods.

We restructured Fig 1 (Fig 2 in the revised manuscript) to clarify its relationship to the simulations and to the main text. We have also rewritten its caption and the corresponding part of the main text.

-Another major point that is not discussed is inhibition. Although little is known about individual inhibitory inputs and their organization, in the author's framework all inhibitory inputs are assumed to be untuned. Given the evidence for tuned inhibitory neurons in the visual cortex, the authors should provide some consideration of how this impacts the conclusions that are drawn.

We note that although we used untuned hippocampal inhibition, our inhibitory inputs in the visual cortex showed weak tuning (Niell & Stryker 2008) as it is described in the Methods. We added a sentence (line 275-77) discussing the role of inhibition in functional synaptic clustering. There are two main possible directions: 1) tuned inhibitory inputs can increase the contribution of input factors, and 2) branch specific inhibitory input can suppress dendritic spikes and thus can decrease the contribution of dendritic factors.

Minor concerns :

- Please check the colors in Fig. S5J for accuracy.

We checked and the color code was correct in Fig. S5J.

- In general, the details of figures should be made less confusing for the reader. For example in Figure 1, it is difficult to figure out what all the letters stand for.

We clarified the meaning of the letters in Fig. 1 (now Fig. 2) in the main text, the caption and also added a legend to panel K. We hope that this will provide sufficient help for the reader to follow the logic of the text and the figure.

- Fig S1F: it is unclear what the colored plots represent

We clarified the description of the colored plots (insets) in the caption.

- Fig S1 D-E: it is not clear what these symbols are supposed to convey.

We added a description of the schematics to the caption.

Reviewer #2 (Remarks to the Author):

The functional impact of synaptic clustering has received a large amount of attention over the years and for good reason. The authors here continue a line of work where they examine the potential impact of dendritic nonlinear synaptic integration (that requires clustering) on the neuronal input output transformations occurring during complex patterns of input in computer models. While this is possibly a fruitful exploration the main difficulties arise from the authors' capability to accurately capture the large number of parameters in play here or in this case to even sufficiently explore the parameter spaces. This paper presents additional simulations that with the current set of parameters show, again, that local dendritic nonlinearities are too weak to heavily impact the simulated response of a single neuron to two different input patterns (theta/ place field like or SPW/ripple like). In the end, this is an empirical question and it is not clear how this manuscript aids in the design and interpretation of experiments meant to provide an answer. Specifics follow.

We thank the reviewer for raising the important point of how our study relates to design and interpretation of experiments. We agree with the reviewer that all theoretical and simulation studies should facilitate the interpretation of existing experimental results and motivate novel experiments. Therefore in the last section in the Discussion we collect the points where we believe that our work contributed to changing the interpretation of existing experimental data and provided specific experimental directions to test the predictions of our theory. Here we repeat the two experiments proposed to directly test our model's predictions.

*“One fundamental prediction is that small clusters of synapses have minimal effect on the response of a neuron under *in vivo* conditions. A direct way to test this prediction is to stimulate a set of inputs of a neuron *in vivo* in clustered and distributed configurations (e.g. by *in vivo* two-photon glutamate uncaging (Noguchi et al. 2011) and compare the resulting somatic response. Another critical insight of our theory is that global plasticity does not account for reinforcement of small coactive synapse clusters. This prediction could be tested by a combination of imaging techniques, whereby one measures the activity of both small functional synapse clusters and the soma (e.g. Iacaruso et al. 2017) and monitors long-term plasticity of the clustered synapses (e.g. Zhang et al. 2015). Specifically, our theory predicts that small clusters of coactive synapses will be strengthened even if they are uncorrelated with somatic activity. While currently both experiments are beyond tractability with available techniques, they could directly test the predictions of our model in the foreseeable future.”*

1) One thing that might help with the impact is for the authors to use their models to explore the parameter spaces in search of models where clustering and the nonlinear dendritic integration hypothesized to detect it would actually impact the firing of a neuron during place fields or ripples.

We would like to note, that we analysed the impact of clustering during sharp waves in searching for input conditions where clustering will actually impact the firing of the neurons.

Actively searching for parameter ranges where small synaptic clusters can impact the firing of neurons can bias our analysis towards over-emphasizing the role of synapse clusters. Our aim was to obtain an unbiased estimate of the effect of synaptic clusters, thus we fitted all the parameters of the model and the inputs before analysing their effect on synapse clustering. Nevertheless, we performed an additional set of analysis, where we doubled all excitatory synaptic conductances and increased inhibition accordingly. Based on our LTP experiments (when only the clustered synapses were doubled), we expected that this manipulation would increase the impact of small clusters. However, as we show it in Figure R2. the impact of clustering was even smaller with stronger inputs, presumably due to the increased conductance load on the cell (similar to the SPW case). We are ready to include this additional analysis in the manuscript, if the reviewer or the editor finds it useful.

Figure R2. Impact of clustering with control parameters (bright colors, same as Fig. 3F) and with stronger synapses (light). The effect was always smaller than with weaker synapses and its maximum was around 10 synapses per cluster.

It springs to mind that the dendritic branch excitability plasticity studied in the past by one of the authors might change the results in figure 3 substantially. What happens if input is clustered on to branches with strongly propagating local Na spikes?

We thank the reviewer for the idea of incorporating high excitability branches. We extended our model to accommodate strong dendritic Na⁺ spikes and studied their influence on the effect of synaptic clustering. Consistent with the previous observation regarding the role of the Na⁺ spikes in dendritic integration (see Losonczy & Magee, 2006), we found that even strong Na⁺ spikes did not significantly alter the slow voltage response of these neurons under in vivo-like conditions and did not change the threshold for nonlinear integration. However, we observed that if these highly excitable dendritic branches receive clustered synaptic input, the spikes can actively propagate from the dendrites to the soma thus increasing the firing rate of the cell. We quantified these effects in Supplementary Figure S4.

2) What is the role of input clustering in synaptic plasticity? The authors focus on what they call global plasticity (repeated pairing of output and input) vs local plasticity (not dependent on output). However, neither of these forms of synaptic potentiation are actually involved in the generation of place fields. Since the potentiation that does underlie place fields requires the overlap in time and space of a locally generated synaptic signal and a large dendritic depolarization (Ca²⁺ plateau) it should be interesting to explore how clustering impacts the generation of the local NMDA dependent signals.

We thank the reviewer for this comment. The plasticity mechanisms involved in the generation of place fields have been under intensive research over the past years. Indeed, a series of recent studies demonstrated that dendritic plateau potentials can transform silent cells into place cells in CA1 in a highly familiar environment (Bittner et al., 2015, 2017). However, it is quite possible that other mechanisms, involving more gradual plasticity mechanisms also contribute to the place field formation under other conditions, e.g. in novel environments (see e.g., Cohen et al., 2017). We added a sentence discussing the potential complementary role of these, and other, local plasticity mechanisms in the generation of the feature selectivity of place cells in the discussion (lines 306-310).

Nevertheless, per the reviewer's suggestion, we examined how clustering impacts the generation of local NMDA dependent signals in hippocampal pyramidal neurons. First, in Fig 4D (Fig 3D in the previous version) we analysed the average actual current flowing through NMDA receptors in the function of synaptic clustering (in the previous version we plotted the maximum possible NMDA current assuming the presence of glutamate). Moreover, we also analysed the frequency of large depolarisation events (mostly dendritic NMDA and Na⁺ spikes) in dendritic branches receiving clustered inputs during both theta and SPW activity, and presented the results of this analysis in Fig 3G and 4G. These analyses confirmed our intuition that in our model NMDA dependent dendritic signals emerge gradually with increasing the size of the synapse clusters.

This will of course require the implementation of dendritic spines with a high resistance necks (which should have been present in all the current simulations and is a good example of how difficult it is to keep track of all the important parameters in a model like this).

To fully address the reviewer's concerns, we rerun all simulations (Figs 2-4 in the revised manuscript) with including dendritic spines with high resistance necks. We found that although including spines slightly changed the neuronal responses, they did not qualitatively affected the results. In our current manuscript all the figures show the data obtained with spines. In Fig 3E-F we also included the data obtained without spines (dashed grey lines) to quantify the contribution of the spines to the neuronal responses. We would like to note, that in our original submission we also included a few simulations with dendritic spines, and based on those data we had anticipated that spines would not qualitatively change the results.

One addition factor to include is the regulation of spine neck resistance to counter spine saturation following LTP (see Harnett lab).

Experimental data suggest that R_{neck} is lowered upon LTP (Tonnesen 2014). In our simulations we used two values for the neck resistance parameter: 500 GOhm or 0 GOhm (corresponding to the absence of spines) matching the wide range of R_{neck} estimates in the literature. The spine neck resistance parameter did not change our results making it unlikely that spine saturation effects or changes in R_{neck} would significantly confound the interpretation of our results.

3) One final point. It would be of interest to have the authors explain what type of information is contained within dendritic clusters in CA1. Perhaps I misread the paper but the authors seem a bit vague about the hypothetical functions of clustering and their detection by dendritic nonlinearities. The old idea has been that this additional layer of processing provided by active dendritic branches allows the neuron to increase the complexity of its feature selectivity by generating higher-order features.

We thank the reviewer for these discussion points. Indeed, our paper is deliberately neutral about the function of synaptic clustering allowing multiple complementary interpretations. As the reviewer suggests, clusters could indeed contribute to increase the flexibility of single neuron representations in CA1, such as having multiple, independent place fields in the same or different environments (Ujfalussy et al., 2009). Alternatively, clusters can be signatures of efficient computation using spiking inputs and presynaptic cell-assemblies (see Ujfalussy et al., 2016). Finally, synaptic clusters could have an important role in triggering various local plasticity rules (Weber et al., 2015, Mago et al., 2019). We briefly clarified these points in the discussion (lines 335-360).

How do the authors see this happening in the hippocampus which is presumably already very high-order even in the CA3?

In our view, the hippocampus implements episodic memory by creating flexible representation of the *configural relationship* between events, objects and locations. While even the representation of the individual elements can be considered as high-order, the configuration of the elements could still require or benefit from additional layers of flexibility, potentially represented by active dendritic processing. Recent theoretical work has begun to explore how the spatial representations described along the entorhino-hippocampal axis can emerge from a more general, relational memory function (Whittington et al., 2019).

In addition, CA3 neurons do not have a fixed feature selectivity (each cell can change its place field location depending on the environmental context) would this cause a problem for the downstream neurons trying to integrate this input?

Indeed, in each context a different CA3 representation will be active, resulting in many possible connectivity configurations activated, depending on the environmental context. When the animal first encounters a context, synapses of the active presynaptic neurons may be initially organised randomly on the postsynaptic dendritic tree of the innervated CA1 neurons. Upon repeated experience of the context, the connectivity can be gradually refined (via spine turnover) by organising co-active synapses into functional clusters. At the end, multiple synapse clusters can coexist in the same postsynaptic neurons, each tuned for a different presynaptic ensemble (Kirchner et al., 2019). It is also possible, that each presynaptic cell (or synapse) participates in multiple functional clusters, but further experimental data is needed to test this possibility. We briefly expanded on these points in the discussion (lines 335-360).

Reviewer #3 (Remarks to the Author):

The paper makes several claims. One is that under conditions designed to be reflective of physiological conditions in hippocampal CA1 pyramidal cells, small randomly occurring synaptic clusters don't influence the somatic membrane potential, in a realistic computer model. Similar results were obtained with a layer 2/3 pyramidal cell model. A second result was that larger synaptic clusters can lead to clustering-based tuning – this was known, but the key point being made here is that this is above the size of synaptic clusters which are observed in vivo. Finally, the third main result is that synaptic clustering has small impact during hippocampal sharp waves.

The first of these results is interesting, although perhaps would not completely surprising. It of course suffers from the problem that it is not possible to prove a negative. Who is to say that there is not some other cell type or physiological condition under which relatively small synaptic clusters may have a larger somatic effect?

We agree with the reviewer that it is possible that small synapse clusters can in principle contribute to neuronal response tuning under different conditions. However, the focus of the model is to study the contribution of synaptic clustering under the typical, physiologically relevant input conditions, where we could show that their contribution is small. Our study could therefore stimulate new experiments looking for physiological states where these small clusters can have a larger impact, new analyses for the more systematic estimates of the cluster size or novel theories about the function of small synapse clusters.

However, the modelling incorporates a lot of details which have been carefully matched to experimental data. I believe that it probably provides a good representation of what is going on in CA1, although it would be nice if this could be validated in some way.

Motivated by the reviewer's comment, we made efforts to further validate our CA1 model. We added five new panels in Figure S1 where we compared several aspects of the somatic membrane potential statistics of our biophysical model to in vivo data (based on Grienberger et al., 2017 Nature Neurosci.) without tuning the model parameters. The good match argues that our model, reproducing not only dendritic integration in vitro but also predicting the neuron's response to in vivo-like stimuli indeed provides a good representation of what is going on in CA1.

I'm not sure how much this is going to influence thinking in the field – this is what most people think anyway. That does not mean that the results should not be published.

To demonstrate the novelty of our manuscript, we collected a few quotes from recent papers where the authors argued that small synaptic clusters can induce dendritic nonlinearities and thus can contribute to the neuronal responses.

Makino & Malinow 2011, discussion: „Clustered plasticity could bind functionally relevant inputs onto dendrites and enhance storage capacity of individual neurons by locally recruiting nonlinear voltage-gated conductances.”

Takahashi et al. 2012 end: “...assemblies are expected to coordinate temporal activity sequences. Such sequential activation would facilitate nonlinear synaptic integration and enhance the computational power of a single neuron.”

Iacaruso et al. 2017 discussion: „Inputs representing similar visual features from overlapping locations in visual space were more likely to terminate on nearby spines, consistent with the idea that co-active inputs cluster on dendritic branches. Neighbouring inputs might cooperate to generate nonlinear dendritic events that contribute to a neuron's output.”

Scholl et al. 2017 discussion: „The prevailing view of input organization on dendritic structures is that clustered inputs amplify functional features contributing to spike generation at the soma and, in the case of the measurements presented here, the RF center” „Our results suggest that synaptic clustering could amplify inputs contributing to both supra- and subthreshold responses and emphasize the challenges that remain in understanding the complex interplay of different factors within the dendritic field that shape a neuron's input/output function. ... A full accounting of the input/output function would need to take into account differences in synaptic strength, the distribution of inhibitory inputs within the dendritic field, as well as the nonlinearities expected to derive from dendritic clustering.”

Fu et al. 2012 discussion: „clustered new spines may synapse with distinct (but presumably functionally related) presynaptic partners. In this case, they could potentially integrate inputs from different neurons nonlinearly and increase the circuit’s computational power.”

Kerlin et al., 2019 discussion: “Diverse behavior-related signals were distributed throughout the dendritic arbor, and were compartmentalized by dendritic distance and branching. This compartmentalization may reflect local dendritic operations that expand the processing and information storage capacity of individual neurons”

Kirchner and Gjorgjieva, 2019: “The transient, precise synchronization of even a small group of synapses can result in the nonlinear summation of synaptic activity, enhancing a neuron’s computational capacity”

These examples demonstrate that the common belief of the field is that small synaptic clusters can trigger dendritic nonlinearities and influence somatic response, which view is challenged by our results.

We agree with the reviewer that most people did not consider the possibility that synaptic clusters can be formed by global plasticity rules. To better emphasize the positive message of our result, we slightly changed the storyline, and emphasize that global plasticity can contribute to the maintenance of synaptic clustering once the clusters have grown large enough to trigger nonlinear dendritic integration presumably due to local plasticity mechanisms.

The second result may help interpret recent experimental data – it suggests that the cluster sizes seen e.g. by Takahashi et al are too small to result in clustering based tuning. I think this is a good point.

We thank the reviewer for the encouragements.

The third result, that clusters have small impact during hippocampal sharp waves, is again negative – it is perhaps worth having in the literature the wide range of circumstances under which synaptic clusters don’t much effect somatic membrane potential, but I’m not sure that is really going to have widespread impact on the neuroscience community.

The effect of dendritic nonlinearities on neuronal responses during various network states has been largely unknown. Experimental studies found that dendritic spikes are particularly prominent during hippocampal sharp waves (Kamondi et al., 1998) but their effect on the neuronal output was previously unknown. Previous modelling work also emphasized the facilitatory role of background activity in generating dendritic spikes (Farinella et al., 2014) without considering how those spikes influence neuronal tuning.

Finally, I should note that the initial section (and supplemental figure S1) to some extent recapitulates previously published work by Cazé et al (Biorxiv, 2015, <https://www.biorxiv.org/content/10.1101/029330v4>). That paper explored the circumstances in which a local learning rule could lead to the formation of functional synaptic clusters, and really should have been cited by the authors.

Similar to earlier works (see e.g., Ujfalussy et al., 2009; Legenstein and Maas, 2011), Caze et al. 2015 studies how local plasticity leads to formation of synaptic clusters and neuronal tuning using simplified neuron models. Our goal is somewhat different here: we aim to determine whether global plasticity rules can also lead to functional synapse clusters. Our learning rule is global - it only depends on the output of the cell but not on the local activation of the dendritic subunit. We did not find how our results would recapitulate Caze et al., 2015, but we included citation of these papers (including the 2017 version of the Caze et al., 2015 manuscript) to support the statement that local plasticity can lead to clustering.

On that note, I am struggling to discover from the paper exactly *how* the cluster formation mechanism proposed here differs from that of Cazé et al, because it has been described very heuristically – I was looking for, in essence, the synaptic plasticity rule. I think that in any revision of this paper, that this should be spelled out much more explicitly.

We included a short description of the plasticity rule in the main text (Eq. 1.) and a more detailed explanation in the Methods section of the paper.

If I have understood correctly, however, the Cazé model already has local plasticity by small synaptic clusters, which I what the current authors are proposing is a better way to go than the “global plasticity” mechanism used here, and which does not seem to be well justified.

Indeed, we conclude that under physiological conditions local plasticity rules are required for the development of synaptic clustering. Our results are stronger than previous statements in two ways 1) whereas others demonstrated that local plasticity can lead to clustering, our results indicate that it is necessary for the initial formation and strengthening of the clusters. 2) we showed that once the clusters are strong enough to trigger dendritic nonlinearities, even global mechanisms can contribute to strengthening synaptic clustering. We clarified these points in the discussion (Local vs. global plasticity, lines 301-302).

Overall, I think that this paper provides some important caveats on over-interpreting some recent experimental results – but I am not sure how much impact these caveats will have on the neuroscience community, because I’m not sure that anyone is interpreting them that way anyway.

As we have detailed above, we perceive a widespread view in the community that the major role of synaptic clustering is to trigger dendritic nonlinearities and influence neuronal tuning. Our paper challenges this view and thus can initiate important discussions in the field to reveal the real impact and function of synapse clusters.

The work is convincing, as far as it pertains to the specific models investigated here – but the level of biophysical detail inherently results in lack of generality.

Experimental studies examining biological systems are also specific to the species, age, brain area or cell type investigated there. We acknowledge that the validity of our current work is limited to the rodent hippocampus during theta and sharp wave state and visual cortex during drifting grating stimulations, but given the number of papers studying these particular systems, we do not find this to be a problem.

I am not sure that this is practically a resolvable issue – perhaps what would strengthen the conclusions more would be accepting that these are investigations of a very specific pair of circumstances (CA1 and L2/3) and attempting experimental validation of these models.

To validate the biophysical model experimentally, we calculated several important measures of the somatic membrane potential in response to the in vivo-like inputs we used in the study and compared it with in vivo intracellular data from navigating animals (Figure S1J-L). Importantly, we performed this comparison with the original biophysical model parameters fitted to in vitro data and input parameters fitted to population activity data but without refining the parameters to reproduce the in vivo single neuron activity data.

Reviewers' Comments:

Reviewer #1:

Remarks to the Author:

The authors have effectively addressed the concerns that were raised in the previous review making this a strong addition to the literature.

Reviewer #2:

Remarks to the Author:

the authors have answered all but one of my questions.

I think it would be highly informative for them to examine the impact of strong dendritic branches on the neuronal output during SPW ripple events in CA1 (not just during theta states as is currently the case). It stands to reason that the only situation in which dendritic Na spike initiation can have an effect on output is during the rather sparse firing that occurs during SPW/ripple events.

Once they have done this analysis this will be a thorough examination of the potential impact of dendritic nonlinearities in CA1.

Reviewer #3:

Remarks to the Author:

The paper is much improved, and I think that the authors have adequately addressed my comments.

A key issue I had previously was lack of clarity on the plasticity rule. The new section around Eq. 1, together with the expanded Methods section on the plasticity model, greatly improves this. I also accept that authors of a number of recent high profile papers have made strong statements about a functional role for small synaptic clusters (even if not necessarily representative of the full weight of opinion in the systems neuroscience community). I think the present study makes a good case for the irrelevance of small synaptic clusters to neuronal functional output, which will at the very least provoke further study to resolve the issue.

Response to reviewers

We thank the reviewers for their constructive comments and support for the paper. Below we address Reviewer #2's concerns (our responses are in italic).

Reviewers' comments:

Reviewer #2 (Remarks to the Author):

The authors have answered all but one of my questions. I think it would be highly informative for them to examine the impact of strong dendritic branches on the neuronal output during SPW ripple events in CA1 (not just during theta states as is currently the case). It stands to reason that the only situation in which dendritic Na spike initiation can have an effect on output is during the rather sparse firing that occurs during SPW/ripple events.

Once they have done this analysis this will be a thorough examination of the potential impact of dendritic nonlinearities in CA1.

We extended our previous analysis regarding the role of Na⁺ spikes initiated in strong dendritic branches on the neuronal output during SPW ripple events in our model CA1 pyramidal neuron. Confirming the Reviewer's intuition, we found that strong dendritic branches have a larger impact on the somatic response during SPWs than during theta. However, in line with the rest of the paper, we found that synaptic clusters had to be large to influence the somatic response through dendritic Na⁺ spikes (Figure 5).

Inspired by the effect of dendritic Na⁺ spikes on the neuronal spiking, we also analysed spike timing during SPWs with and without strong dendritic branches. We found that the replayed trajectory is encoded by the timing of the first action potential during a SPW but we did not find evidence for the modulation of this temporal code by the presence of strong branches (Supplementary Fig. 4).

Reviewers' Comments:

Reviewer #2:

Remarks to the Author:

In my opinion, this manuscript is complete now. Its good work.

Response to reviewers

Reviewers' comments:

Reviewer #2 (Remarks to the Author):

In my opinion, this manuscript is complete now. Its good work.

We thank the reviewer for his constructive comments and support for the manuscript throughout the review process.